# Cellular composition and circuit organization of the locus coeruleus of adult mice

Andrew McKinney[1,2,3], Ming Hu[2,3], Amber Hoskins[4], Arian Mohammadyar[4], Nabeeha Naeem[4], Junzhan Jing[2,3], Saumil S Patel[2], Bhavin R Sheth[5,6]*, Xiaolong Jiang[1,2,3,7]*

[1]Neuroscience Graduate Program, Baylor College of Medicine, Houston, United States; [2]Department of Neuroscience, Baylor College of Medicine, Houston, United States; [3]Jan and Dan Duncan Neurological Research Institute at Texas Children's Hospital, Houston, United States; [4]University of Houston, Houston, United States; [5]Department of Electrical and Computer Engineering, University of Houston, Houston, United States; [6]Center for NeuroEngineering and Cognitive Science, University of Houston, Houston, United States; [7]Department of Ophthalmology, Baylor College of Medicine, Houston, United States

**Abstract** The locus coeruleus (LC) houses the vast majority of noradrenergic neurons in the brain and regulates many fundamental functions, including fight and flight response, attention control, and sleep/wake cycles. While efferent projections of the LC have been extensively investigated, little is known about its local circuit organization. Here, we performed large-scale multipatch recordings of noradrenergic neurons in adult mouse LC to profile their morpho-electric properties while simultaneously examining their interactions. LC noradrenergic neurons are diverse and could be classified into two major morpho-electric types. While fast excitatory synaptic transmission among LC noradrenergic neurons was not observed in our preparation, these mature LC neurons connected via gap junction at a rate similar to their early developmental stage and comparable to other brain regions. Most electrical connections form between dendrites and are restricted to narrowly spaced pairs or small clusters of neurons of the same type. In addition, more than two electrically coupled cell pairs were often identified across a cohort of neurons from individual multicell recording sets that followed a chain-like organizational pattern. The assembly of LC noradrenergic neurons thus follows a spatial and cell-type-specific wiring principle that may be imposed by a unique chain-like rule.

*For correspondence:
brsheth@uh.edu (BRS);
xiaolonj@bcm.edu (XJ)

Competing interest: The authors declare that no competing interests exist.

## Editor's evaluation

Recent studies of the brainstem locus coeruleus (LC) noradrenaline system have demonstrated a modular functional organization, yet how different noradrenaline cell classes are independently regulated is not clear. Using ex-vivo, multi-patch recordings of up to eight LC neurons at once, this study offers compelling evidence for the existence of two morpho-electric cell classes with segregated electrical coupling. These important findings establish principles of local circuit communication occurring preferentially within defined LC-noradrenaline cell classes.

## Introduction

The locus coeruleus (LC) is a small pontine nucleus that houses the vast majority of norepinephrine (NE)-producing neurons in the brain and sends widespread noradrenergic projections to all major

divisions of the central nervous system (CNS) (*Schwarz and Luo, 2015*). By releasing NE across the CNS, the LC participates in many cognitive processes and behaviors crucial for survival, including fight and flight response, attentional control, and sleep/wake cycles, while its dysregulation is implicated in various neuropsychiatric conditions, including depression, anxiety, autism spectrum disorder, and post-traumatic stress disorder (*Bast et al., 2018*; *Fortress et al., 2015*; *Southwick et al., 1999*; *Weinshenker, 2018*). Given its small size, the LC has been long considered as a homogeneous cluster of NE-producing neurons (LC/NE neurons) exerting global, uniform influence over all CNS divisions (*Sara and Bouret, 2012*; *Schwarz and Luo, 2015*). However, increasing evidence supports LC/NE neurons are heterogeneous in terms of the anatomical projections, morpho-electric profiles, and specific behavioral functions they involve (*Bari et al., 2020*; *Hirschberg et al., 2017*; *Li et al., 2016*). Particularly, recent tracing studies coupled with behavioral paradigms suggest that the LC is composed of multiple specialized noradrenergic modules. Each module comprises nonoverlapping LC/NE neurons with specific electrophysiological properties and projection targets, thus participating in distinct functions of the circuits within its efferent domain (*Chandler et al., 2014*; *Li et al., 2016*; *Totah et al., 2019*; *Uematsu et al., 2017*). Interestingly, LC/NE neurons within each module are also able to switch from a discrete, patterned coding mode to a global broadcast mode to cope with the more demanding situation (*Uematsu et al., 2017*). Such functional modularity in the LC and their context-dependent adaptivity suggest LC neurons are functionally heterogeneous and may interact with each other in a complex and subtle way within the local circuit to coordinate the division of labor. To understand how each LC module is functionally specialized and how distinct LC modules interact to achieve specific functions, it is essential to dissect the local circuit organization of the LC in terms of its cellular diversity and wiring logic.

LC/NE neurons belong to monoaminergic systems, and their axon terminals are believed to be 'asynaptic'-free nerve endings from which NE is released to modulate the activity of nearby fast-acting synapses releasing glutamate or GABA in the form of 'volume transmission.' Increasing evidence suggests monoamine neurons, including LC/NE neurons, could co-release glutamate to engage in fast neurotransmission as well (*Fung et al., 1994a*; *Fung et al., 1994b*; *Yang et al., 2021*). It appears that NE can be released locally in the LC to activate α2 adrenoceptor (α2AR) on the same or adjacent LC/NE neurons to initiate a negative feedback mechanism regulating their excitability (*Aghajanian and VanderMaelen, 1982*; *Egan et al., 1983*; *Huang et al., 2007*; *Williams et al., 1985*), but it remains to be illustrated whether these neurons can interact with each other via fast neurotransmission. In addition, it is long believed that LC/NE neurons form extensive gap junctions to synchronize their activity as a global broadcast. However, gap junctions among LC/NE neurons appear to be prevalent only in neonates and decline significantly with age (*Christie and Jelinek, 1993*; *Christie et al., 1989*; *Travagli et al., 1995*). It remains to be determined with direct demonstration if electrical coupling persists within mature LC/NE neurons to mediate their synchronization (*Ishimatsu and Williams, 1996*; *Patrone et al., 2014*; *Rash et al., 2007*; *Travagli et al., 1995*).

To answer all these questions, here we adopted several strategies for studying cortical circuits, including adult tissue slicing and simultaneous multipatch recording (*Jiang et al., 2015*; *Ting et al., 2014*), to the brainstem to interrogate LC local circuit in adult mice. By profiling the morpho-electric properties of each LC/NE neuron and examining their connectivity at an unprecedented scale and level of detail (>700 neurons), we have uncovered and defined two major morpho-electric noradrenergic cell types in the LC and deciphered the major mechanism governing their interactions at single-cell resolution. Our results indicate that at least in our horizontal slice preparation, no glutamatergic excitatory transmission was detected among LC neurons. Instead, we found electrical coupling is the major synaptic mechanism by which mature LC neurons communicate with each other, and their assembly follows a spatial and cell-type-specific wiring principle that may be imposed by a unique chain-like rule.

## Results

### Electrophysiological and morphological heterogeneity of LC/NE neurons

Each brainstem slice prepared from adult mice with an NMDG-based slicing protocol contained numerous healthy LC neurons (*Jiang et al., 2015*; *Kuo et al., 2020*; *Ting et al., 2014*), allowing for

simultaneous patch-clamp recordings of up to eight mature LC neurons (*Figure 1A*). To ensure only noradrenergic neurons (henceforth LC/NE neurons) were studied (i.e., surrounding GABAergic neurons in the peri-LC area were excluded) (*Breton-Provencher and Sur, 2019*), only fluorescence-positive neurons were targeted for recordings in each slice prepared from *Dbh*-Cre: Ai9 mice (*Figure 1A*). To put their electrophysiology in a broader context of slice electrophysiology, we also recorded pyramidal cells (CA1-PCs) and parvalbumin-expressing interneurons (CA1-PVs) from the hippocampal CA1, two cell types commonly recorded and studied in brain slices, using the same conditions. LC/NE neurons at rest had a highly depolarized resting membrane potential (RMP, –43.0 ± 0.1 mV, n=289), and all neurons were spontaneously active at rest (1.6 ± 0.4 Hz, n=30; *Figure 1B*), similar to previous reports on LC neurons (*Andrzejewski et al., 2001*; *Ballantyne et al., 2004*; *Kuo et al., 2020*), but different from hippocampal neurons (*Figure 1B*). In response to negative current injection, LC/NE neurons were charged much more slowly with a higher input resistance compared to both types of hippocampal neurons (*Figure 1C*). In response to prolonged suprathreshold positive current injection, the majority of LC/NE neurons exhibited a characteristic pause after their first action potentials (APs) followed by a series of APs, which departed significantly from stereotypical firing patterns of hippocampal neurons (*Figure 1C*). LC/NE neurons were also unique in their single APs, which were much wider with a lower amplitude, followed by a deep and prolonged afterhyperpolarization compared to hippocampal neurons (*Figure 1C and D*). To illustrate strikingly distinct properties of LC/NE neurons, we extracted 13 electrophysiological features from three groups of neurons (see 'Materials and methods') and plotted them in a t-SNE-map. This t-SNE-based clustering approach clearly separated LC neurons from the two types of hippocampal neurons (*Figure 1D*), highlighting the unique electrophysiological properties of LC neurons.

We recovered the morphology of each recorded LC neuron to characterize their morphological features (*Figure 2A and C*, *Figure 2—figure supplement 1*). Most LC neurons were large cells with smooth dendrites (aspiny), similar to CA1-PVs in the hippocampus, but different from CA1-PCs with spiny dendrites (*Figure 2—figure supplement 1*). In addition, their dendritic trees were more tortuous with less branch order than hippocampal neurons (*Figure 2—figure supplement 1*). LC/NE neurons had diverse somatodendritic shapes, based on which we could group them into two major types: fusiform cells (FF) and multipolar cells (MP) following previous nomenclature (*Swanson, 1976*). FFs had an elongated, usually spindle-shaped large soma with two, or rarely three, primary dendrites originating from opposite poles of the soma (*Figure 2C, D and F*). The second type of neurons, MP, had a triangular or nearly round soma with more than three primary dendrites that could originate from any side of their soma (*Figure 2A, B and F*). The dendrites of MPs also branched more profusely around the soma compared to that of FF (*Figure 2A, B and G*, *Figure 2—figure supplement 2*). To support LC neuron classification, we reconstructed a subset of LC/NE neurons and quantified their soma bipolarity (i.e., the ratio of the major to the minor axis of a fitted ellipse of the soma; the larger the ratio, the more bipolarity the soma exhibits) and the bipolarity index of dendritic arborization (the proportion of the dendrites originating around the major axis of the fitted ellipse; the larger the proportion, the more bipolarity the dendritic arborizations exhibits) (*Figure 2E*), and then plotted these two parameters against each other. Two types of neurons were well separated in this plot (*Figure 2E*). We also trained a nonlinear neural network-based classifier to distinguish cell types and the classifier could separate two types with a performance accuracy of 100% (*Figure 2—figure supplement 3C*). Of note, the performance using soma shape alone was almost as good as the performance using two parameters (*Figure 2—figure supplement 3A and B*), suggesting that soma shape is a reliable criterion to distinguish two types of neurons. These analyses supported our manual classification based on the somatodendritic shape.

In addition to somatodendritic differences, MP and FF differed in their axonal arborization as well. The majority of LC/NE neurons had a visible thick axon that originated from either the soma or dendrite (see below), and then arborized away from the fourth ventricle and were inevitably severed due to the slicing procedure (as evidenced by a characteristic retraction ball) (*Blythe et al., 2009*). FFs generally had the axon originating from one of the primary dendrites (~95%, 37 out of 39 reconstructed FF neurons, *Figure 2D*, *Figure 2—figure supplement 2B*), suggesting that most FFs have privileged dendrites (i.e., synapses made on these axon-bearing dendrites are in an electrotonically privileged position in evoking action potentials)(*Häusser et al., 1995*), while most MPs had their axon originating from the soma (~80%, 37 out of 46 reconstructed MPs; *Figure 2B*, *Figure 2—figure supplement 2A*).

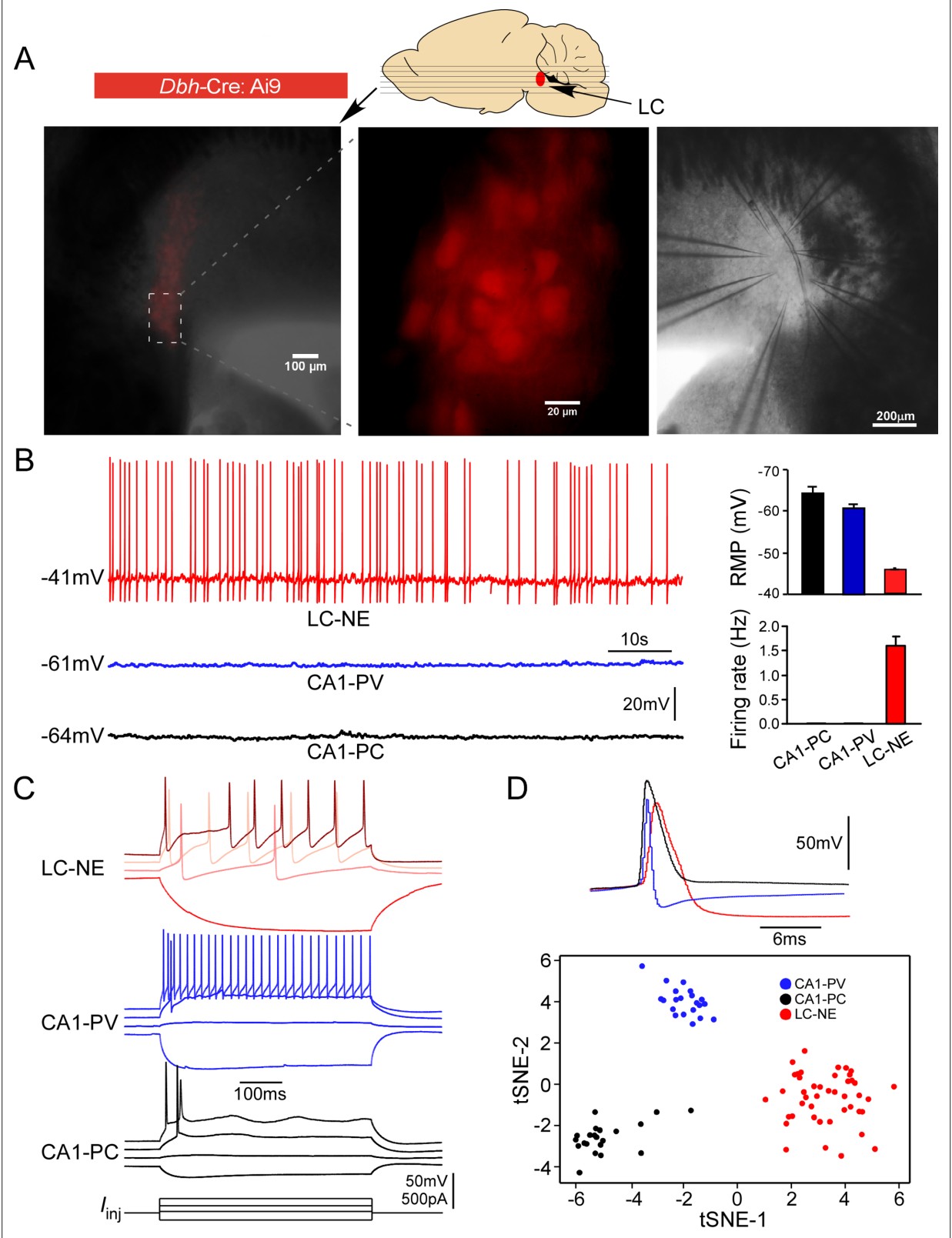

**Figure 1.** Electrophysiological properties of locus coeruleus/norepinephrine (LC/NE) neurons. (**A**) Horizontal slices prepared from *Dbh*-cre:Ai9 mice for multicell patch-clamp recordings of LC/NE neurons. (**B**) Spontaneous firing of LC-NE neurons at rest and their more depolarized resting membrane potential (RMP) compared to hippocampal CA1 pyramidal cells (PC) and parvalbumin-expressing interneurons (PV). The representative trace for each cell type at rest is shown on the left. Right: average RMP (top, n=20 for CA1-PC and CA1-PV, n=289 for LC/NE, CA1-PC vs. LC/NE, p<0.0001; CA1-PV vs.

Figure 1 continued

LC/NE, p<0.0001) and average firing frequency (bottom, n=20 for CA1-PC and for CA1-PV; n=30 for LC/NE; CA1-PC vs. LC/NE, p<0.001; CA1-PV vs. LC/NE, p<0.001) of LC/NE neurons compared to hippocampal CA1-PC and CA-PV neurons. (C) The membrane potential response to the step stimulation protocol (10 pA incremental) of LC/NE neurons compared to CA1-PC and CA1-PV neurons. Scale bar, vertical 50 mV for the potential, 500 pA for the injected currents. (D) Individual action potential (AP) of three types of neurons (top) and tSNE of their intrinsic electrophysiology properties (bottom).

In addition, MPs had less local axonal arborization as compared to FFs (*Figure 2G*, *Figure 2—figure supplement 2*). The axon in many MPs had the first branching point located >400 µm away from their origin, while the axon in FFs branched much earlier and formed denser local axonal arborization than MPs (*Figure 2G*, *Figure 2—figure supplement 2*).

FFs and MPs had different population sizes (MP: ~62%, 416 out of 666 recovered LC/NE neurons; FF: ~38%, 250 out of 666 LC/NE neurons). Given that FFs constitute ~38% of total LC/NE neurons, the proportion of FFs in each multicell recording set (the cohort of >3 LCs neurons from each slice) should fall most frequently within the range of 30–40% if FFs are evenly distributed across the entire LC. However, this was not the case in the histogram of FF proportion (*Figure 2—figure supplement 4*) across all recording sets. Instead, the FF proportions were heavily biased toward either the range of 0–20% or of 50–100%, suggesting that FFs were not evenly distributed across the entire LC, but preferentially localized in certain anatomical positions. Indeed, we often recorded a cohort of neurons in which most LC/NE neurons were FFs (*Figure 2C*). The same was true for the proportions of MP (*Figure 2A*, *Figure 2—figure supplement 4*). This sampling bias might reflect the anatomical preference of different types of LC/NE neurons as reported in rat LC (*Loughlin et al., 1986*; *Swanson, 1976*).

To examine whether MPs and FFs had different intrinsic membrane properties and firing patterns, we extracted and compared eight electrophysiological parameters between the two types (*Figure 2—source data 1*). FFs had a narrower spike with a larger amplitude than MPs (*Figure 2I*, *Figure 2—source data 1*). The spikes from the two cell types also had different temporal dynamics (measured at the first spike evoked by our step protocol, *Figure 2I*, *Figure 2—source data 1*). In addition, FFs could be evoked more spikes with large suprathreshold depolarizing current injections than MPs (*Figure 2H*).

## Synaptic transmission between LC/NE neurons

LC/NE neurons may co-release glutamate to engage in fast excitatory synaptic transmissions (*Fung et al., 1994a*; *Fung et al., 1994b*; *Yang et al., 2021*). To directly test this connection scenario, we examined synaptic connections between any LC/NE cell pair once simultaneous multi-cell whole-cell recordings were established (*Figure 3A and B*). Single APs or a train of APs (5 APs at 20 Hz) were evoked in presynaptic neurons, while membrane potentials of other simultaneously recorded LC neurons were monitored. An excitatory synaptic connection is identified if the excitatory postsynaptic potentials (EPSPs) in potential postsynaptic neurons are time-locked to presynaptic APs (*Jiang et al., 2015*; *Scala et al., 2019*). We tested more than 1500 LC neuron pairs, most of which have an inter-soma distance <150 µm (*Figure 3C*), but no single excitatory synaptic connection was detected (*Figure 3D*) despite prominent spontaneous EPSPs present in each potential postsynaptic neuron. In addition, NE released from LC/NE neurons may activate α2AR on adjacent LC/NE neurons or themselves to evoke hyperpolarization (*Williams et al., 1985*). However, no visible hyperpolarization in potential postsynaptic cells following single presynaptic APs or a train of APs was detected in these tested pairs as well, suggesting NE released from single LC neurons was not sufficient enough to activate the α2ARs.

With our multipatch recordings, we also examined electrical coupling between any cell pair and identified sparse electrical connections among LC/NE neurons at the adult stage (*Figure 3D*). The presence of electrical coupling was tested by recording membrane responses in simultaneously recorded LC/NE neurons following the injection of prolonged depolarizing or hyperpolarizing current pulses (600 ms) in one of the cells. An example of a coupled pair is illustrated in *Figure 4A–C*, in which a current pulse in presynaptic Cell 1 evoked a membrane response of the same sign in postsynaptic Cell 4, albeit of much lower amplitude. The same occurred when a current pulse was injected into Cell 4. Other tested pairs did not show evidence of electrical coupling in either direction (*Figure 4C*).

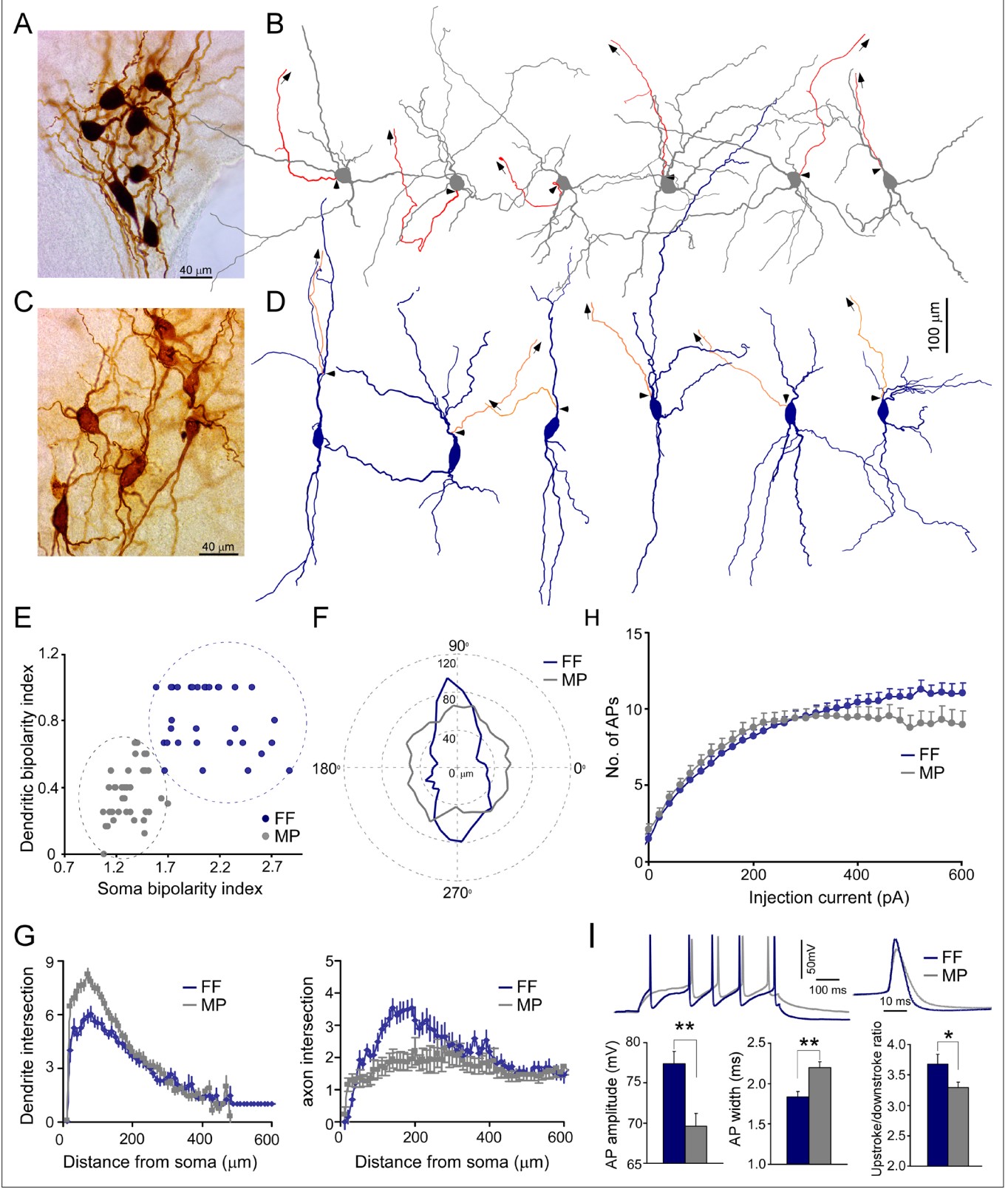

**Figure 2.** Two morphological cell types of locus coeruleus/norepinephrine (LC/NE) neurons. (**A**) A micrograph shows seven morphologically recovered LC/NE neurons, most of which are multipolar cells (MP). (**B**) Morphological reconstruction of six MPs. The dendrite and soma in gray and the axon in red. The axon is shown only partially, and the arrows indicate the trajectory of the major axonal branch. The triangles indicate the origin of the axon. (**C**) A micrograph shows seven morphologically recovered LC/NE neurons, most of which are fusiform cells (FF). (**D**) Morphological reconstruction of six

*Figure 2 continued on next page*

*Figure 2 continued*

FFs. The dendrite and soma in dark blue, and the axon in orange. The axon is shown only partially, and the arrows indicate the trajectory of the major axonal branch. The triangles indicate the origin of the axon. (**E**) Plotting the bipolarity index of the soma (x-axis) against the bipolarity index of dendritic arborization (y-axis; n=31 for FF and n=39 for MP). (**F**) The dendritic length as a function of dendritic orientation of FF and MP. Each radius represents the dendritic length from the soma center at a certain degree. Each morphology is rotated to ensure its major soma axis aligns with the x-axis (90–270°) in the 2D space. Zero in the plot indicates the center of the soma (n=31 for FF, n=39 for MP). (**G**) Left: Sholl analysis of dendritic arbors of MP and FF (counting the number of dendritic intersections that occur at fixed distances from the soma in concentric circles). FF vs. MP: F(1) = 132.9; p<0.0001 with two-way ANOVA, n=32 for FF and n=36 for MP. Right: Sholl analysis of axonal arbors of MP and FF. FF vs. MP: F(1) = 87.3; p<0.0001, n=25 for FF and n=31 for MP. (**H**) The membrane responses (the number of action potentials (APs) evoked) to increasing current step injections in FF and MP (FF: n=39; MP, n=37; F(1) = 29.5; *p<0.001 with two-way ANOVA). (**I**) Electrophysiological differences between cell types. Top: two representative firing traces in response to suprathreshold depolarizing current injections into MP and FF (left), and individual AP from MP and FF (right). Bottom: average AP amplitude (left; FF, n=19; MP, n=41; **p<0.01), average AP half-width (middle; FF, n=19; MP, n=41; **p<0.01), and upstroke/downstroke ratio of AP (right; FF, n=19; MP, n=41; *p<0.05). Also see *Figure 2—figure supplements 1–4* and *Figure 2—source data 1*.

The online version of this article includes the following source data and figure supplement(s) for figure 2:

**Source data 1.** Electrophysiological features of FF and MP.

**Figure supplement 1.** Dendrites of locus coeruleus/norepinephrine (LC/NE) neurons compared to CA1 hippocampal pyramidal neurons (CA1-PCs) and PV-expressing interneuron (CA1-PV).

**Figure supplement 2.** Comparison of axonal and dendritic arbors of two locus coeruleus/norepinephrine (LC/NE) morphological cell types.

**Figure supplement 3.** Machine learning to distinguish two locus coeruleus/norepinephrine (LC/NE) morphological cell types.

**Figure supplement 4.** The proportion of FF or MP across all individual recording sets (slices).

The steady-state coupling coefficient, defined as the voltage change at the postsynaptic cell divided by the voltage change at the presynaptic cell measured at the last 10–20 ms period of the current step (see 'Materials and methods'), was calculated for each recorded pair. A pair of LC/NE neurons was considered to be electrically coupled when the steady-state coupling coefficient was ≥0.005, a criterion determined by the noise of the recording conditions (*Curti et al., 2012*). According to this criterion, 65 of the 1503 tested pairs across the inter-soma distance range of 16–531 μm were electrically coupled (4.3%; *Figure 3D*, *Figure 4F*; also see below). For each coupled pair, the coupling coefficients were estimated and expressed as the average of the values in both directions. The steady-state coupling coefficient averaged 0.040±0.028 (SD) (range: 0.0051–0.13; n=65) (*Figure 4D and I*). The strength of electrical transmission in the vast majority of coupled pairs was bidirectional and symmetric (*Figure 4C and D*). Estimates of the coupling coefficient in both directions for each pair showed a positive correlation with a slope of 1.03 (*Figure 4D*), not significantly different from 1 (p=0.8; *Figure 4D*), indicating that the conductance of putative gap junctions between LC/NE neurons was largely nonrectifying.

Electrical coupling generally occurs between narrowly spaced cell pairs. Given the wide range of the inter-soma distance of tested pairs, a low coupling rate in our dataset was not unexpected. To examine how electrical coupling among LC neurons was restrained by their inter-soma distance, we measured the inter-soma distances of all tested cell pairs to derive their electrical coupling rate as a function of the inter-soma distance. With an inter-soma distance <40 μm, the coupling rate between LC neurons was ~12%, indicating that electrical connections were prominent among LC neurons in adult mice. However, the coupling rate dropped quickly as the inter-soma distance increased from 40 μm (*Figure 4E*). Interestingly, while the coupling rate was much lower in cell pairs with the inter-soma distance >40 μm, it was much more resistant to the increase of the inter-soma distance than the rate in cell pairs with the inter-soma distance <40 μm (*Figure 4E*). The coupling rate remained relatively constant when the inter-soma distance increased from 40 μm to 240 μm before dropping to zero, with a second small peak around 140 μm (*Figure 4E*).

Previous reports suggest that electrical connections among LC/NE neurons decline significantly with age (*Christie and Jelinek, 1993*; *Christie et al., 1989*; *Travagli et al., 1995*). To examine whether this is the case, we performed additional multi-cell recordings of LC neurons from mice at the age of PND 13–21 (*Figure 4F*). Surprisingly, the coupling rate at this young age was very similar to the rate found in adults (4.8%; 8 connections out of 165, p>0.05, *Figure 4F*). The coupling rate was still very similar between the two ages when the inter-soma distance was controlled (*Figure 4F*).

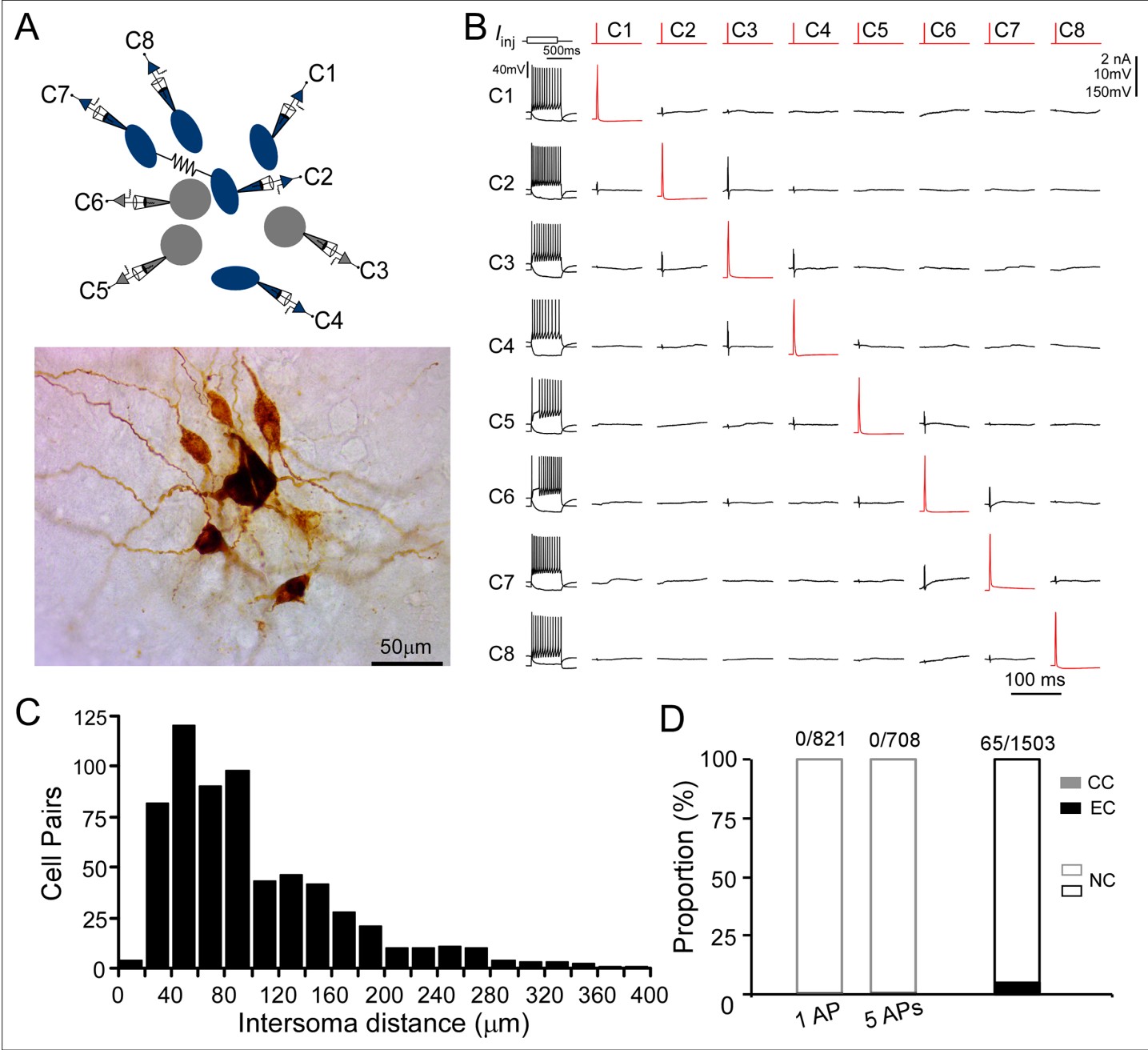

**Figure 3.** Chemical and electrical transmission between locus coeruleus/norepinephrine (LC/NE) neurons. (**A**) Simultaneous patch-clamp recording of eight LC neurons with their morphology being post hoc recovered. Cell 7 and Cell 2 are electrically coupled. (**B**) The membrane responses of eight LC neurons to a hyperpolarizing current injection (bottom trace) and suprathreshold depolarizing current injection (upper trace) and pairwise testing of synaptic connection (or any other type of interactions) between eight LC neurons. Red traces indicate action potential (AP) evoked in each presynaptic cell and the average traces of the potential postsynaptic responses are shown in black. No visible membrane hyperpolarization or depolarization following each AP was detected in any cell pair. Vertical scale bar: 2nA for current injection, 10 mV for the traces in postsynaptic cells, 150 mV for AP evoked in presynaptic cells. The traces for electrical coupling not shown. (**C**) Inter-soma distance of all tested cell pairs. The majority of cell pairs have an inter-soma distance <150 μm. (**D**) The proportion of chemical connections (CC), electrical connections (EC), and non-connection (NC) out of all tested connection pairs. No chemical connection was detected with single AP or a train of APs (5 APs at 20 Hz) across all tested pairs, but electrical connections were sparsely identified in a small subset of cell pairs.

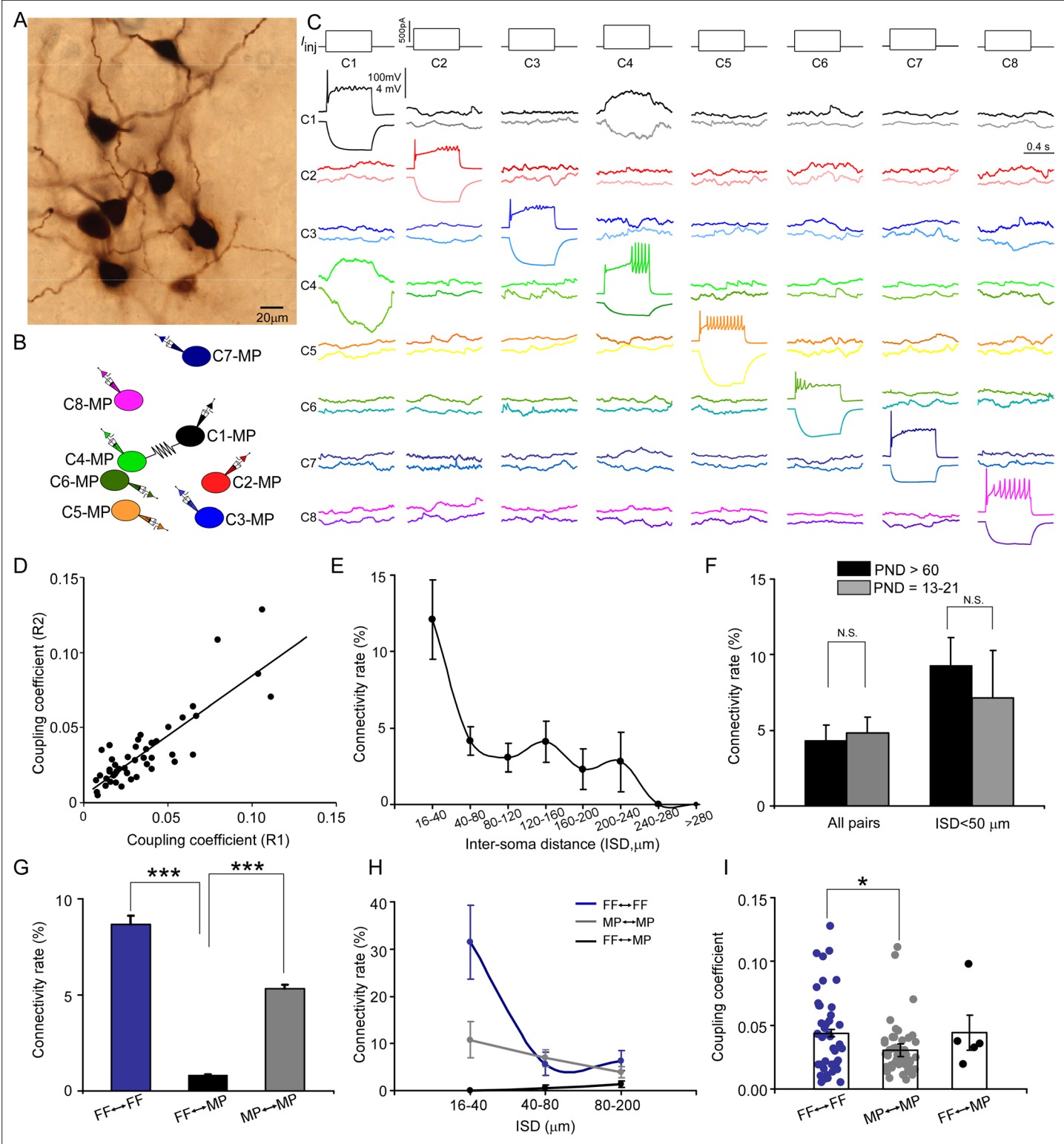

**Figure 4.** Electrical coupling between locus coeruleus/norepinephrine (LC/NE) neurons. (**A**) A micrograph shows eight morphologically recovered LC/NE neurons. (**B**) The relative position of eight neurons (with an assigned number) and their connections. (**C**) Pairwise testing between eight cells as shown in (**A**) reveals electrical coupling between Cell 1 and Cell 4. No sign of electrical coupling between the rest of cell pairs. Vertical scale bar, 4 mV for the potential traces with no current injections, 100 mV for the traces evoked by injected currents. (**D**) Estimates of the coupling coefficient in both directions (R1 and R2) are plotted against each other, indicating that most contacts lack significant rectification. The data are fitted with a straight-line function ($R^2$=0.75, p<0.001). (**E**) Electrical connectivity rate as a function of the inter-soma distance (ISD). (**F**) Electrical connectivity rate at different ages

*Figure 4 continued on next page*

*Figure 4 continued*

(PND: postnatal day). N.S.: no significant difference. (**G**) Electrical connectivity rate between neurons of same type or different types across all cell pairs. ***p<0.0001. (**H**) Electrical connectivity rate as a function of ISD imposed by cell types. (**I**) Coupling coefficients in two homotypic types of electrical coupling (within-cell type) and one heterotypic type of electrical coupling (among the different cell types). *p<0.05.

## Electrical transmission by morphological cell type in LC

Electrical coupling generally occurs between cell pairs of the same cell type in many brain regions (*Bennett and Zukin, 2004*; *Hidaka et al., 2004*). To examine whether electrical coupling between LC/NE neurons was also cell-type specific, we examined all tested cell pairs with recovered morphology and assigned each cell to either FF or MP, and then derived the coupling rates as a function of cell type (*Figure 3*, *Figure 4*, *Figure 5* and *Figure 6*). Of all FF–FF pairs (i.e., both cells are FF), we found 8.7% were electrically coupled (*Figure 4G*); of all MP–MP pairs (i.e., both cells are MP), 5.3% were electrically coupled (*Figure 4G*). By contrast, only 0.8% of FF–MP pairs were coupled (*Figure 4G*; p<0.0001 for both comparisons). While the coupling rate of FF–FF pairs was slightly higher than that of MP–MP pairs, there was no significant difference between these two (p=0.08). Given that the coupling rate is sensitive to the inter-soma distance, we then compared their rates within the same inter-soma distance range to rule out the possibility that the rate differences we observed were due to the differences in the inter-soma distance. Interestingly, with the inter-soma distance of up to 40 μm, no connection was identified in FF–MP pairs (0 out of 55 tested pairs). In contrast, the coupling rates for FF–FF and MP–MP pairs were much higher (11 out of 35 tested pairs, 31.4% for FF–FF, p<0.01 compared with FF–MP; 7 out of 65, 10.7% for MP–MP, p<0.01 compared with FF–MP; *Figure 4H*). In addition, the coupling rate for FF–FF pairs was significantly higher than that for MP–MP pairs over this inter-soma distance range (p=0.01, *Figure 4H*). With the inter-soma distance range of 40–80 μm, the coupling rates for both MP–MP and FF–FF pairs were also significantly higher than that for FF–MP pairs (p<0.01 for both comparisons, *Figure 4H*), but there was no significant difference between the MP–MP coupling rate and FF–FF coupling rate (p=0.67). Electrical coupling in FF–MP pairs most occurred in those cell pairs with the inter-soma distance >80 μm, but their rate even in this inter-soma distance range was lower than the rates for MP–MP and FF–FF pairs (FF–FF vs. FF–MP: 6.3% vs. 1.4%, p<0.01; MP–MP vs. FF–MP: 3.9% vs. 1.4%, p=0.05, *Figure 4H*). These results indicate that electrical connections in LC neurons generally form among cell pairs of the same cell type (*Figures 3 and 4*, *Figures 5 and 6*). Within the same cell type, an FF is more likely to connect with another FF than an MP connects with another MP, particularly if the neurons are located nearby. In addition, electrical coupling in FF–FF pairs had higher connection strength (i.e., higher coupling coefficients) than electrical coupling in MP–MP pairs (*Figure 4I*).

## Putative electrical synapses in LC/NE neurons

We reason electrical synapses formed between FFs (FF–FF) and those formed between MPs (MP–MP) may locate in different somatodendritic domains (soma vs. dendrite), which may contribute to their coupling coefficient difference (*Figure 4I*). To test this hypothesis, we took advantage of high-quality morphology recovered from each LC neuron (*Figures 2–6*) to examine putative electrical synapses between any coupled pairs. We traced the entire somatodendritic domain of a coupled pair under a ×100 oil immersion objective lens and scrutinized those areas where two neurons have overlapping somatodendritic structures. In those coupled cell pairs, a dendritic branch from one cell merged with a dendritic segment from another cell at one or multiple crossing points (*Figure 5A–C*), whereas we rarely found these close contacts in uncoupled cell pairs. We thus believed that these contacts were putative electrical synapses among coupled pairs.

For many coupled pairs, only one putative synapse could be identified (*Figure 5C*). More than two putative synaptic contacts could be identified in some coupled pairs as well (*Figure 5A*) – these were more frequent in FF–FF pairs (*Figure 5G*). This may be one reason for a higher connection strength in FF–FF coupled pairs as compared to MP–MP coupled pairs (*Figure 4I*). Indeed, putative synaptic numbers and coupling coefficients showed a positive correlation (r=0.86, *Figure 5E*). The vast majority of putative synapses were found to form between dendrites: this could be a single crossing point between two dendrites from a coupled pair (*Figure 5C*), or multiple merging sites between an entire dendrite from one cell and the entire dendrite from another cell (in parallel with each other along their course to the terminals; *Figure 5A*). Only two coupled pairs (one FF–FF and one MP–MP) had

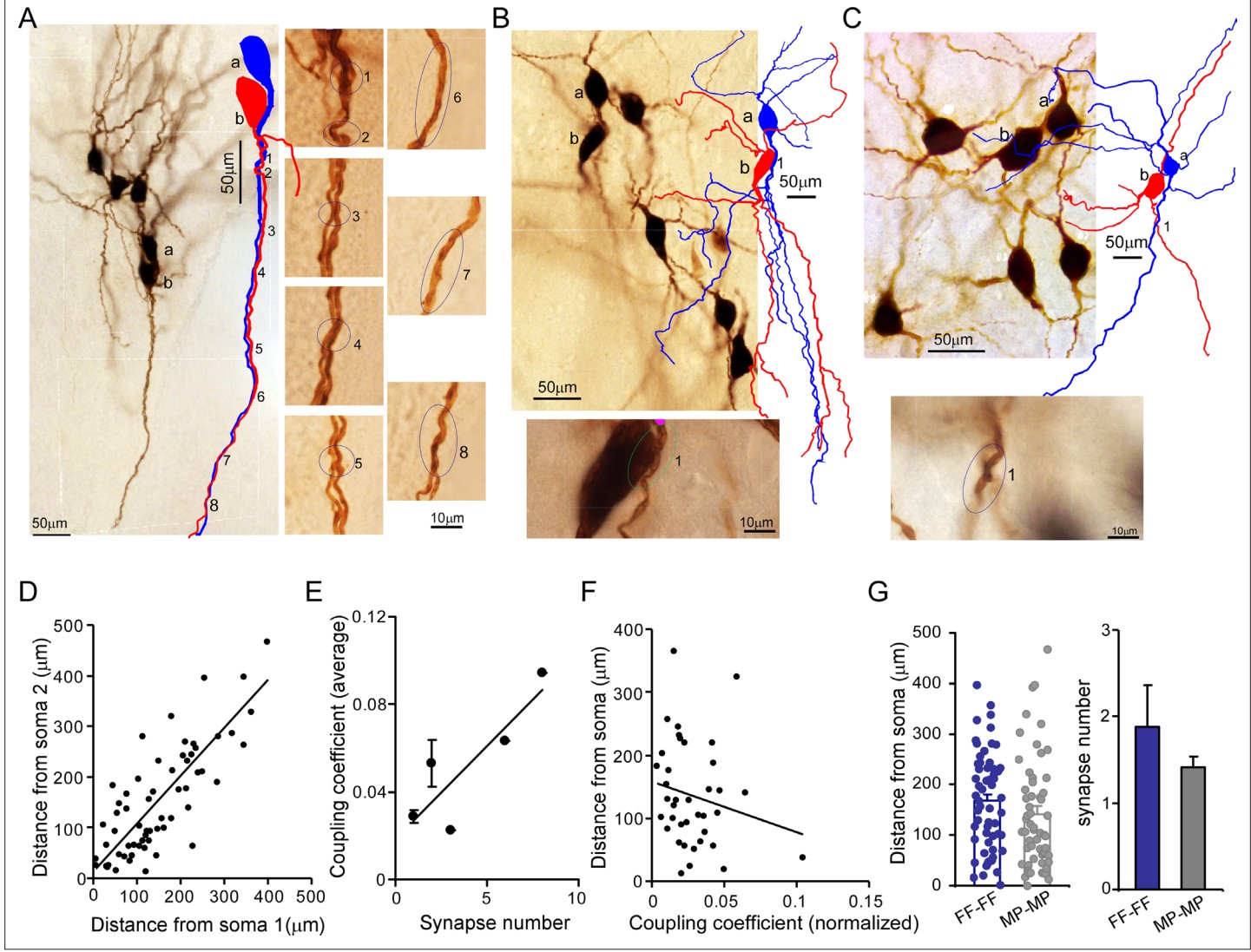

**Figure 5.** Putative electrical synapses in locus coeruleus/norepinephrine (LC/NE) neurons. (**A**) Left: a microphotograph depicting six morphologically recovered LC/NE neurons (simultaneously recorded), among which Cell a and Cell b were identified as an electrically coupled pair. Multiple putative dendrodendritic electrical synapses can be identified after their morphology was reconstructed (as labeled 1 from 8). Right: a high-magnification view of each putative electrical synapse as shown on the left. (**B**) Top: a microphotograph depicting seven morphologically recovered LC/NE neurons, among which Cell a and Cell b were identified as a electrically coupled pair. A putative somatodendritic electrical synapse was identified after their morphology was reconstructed (as labeled as 1). Bottom: a high-magnification view of the putative electrical synapse as shown on the top. (**C**) Top: a microphotograph depicting six morphologically recovered LC/NE neurons, among which Cell a and Cell b were identified as a electrically coupled pair. A putative dendrodendritic electrical synapse was identified (as labeled as 1). Bottom: a high-magnification view of this putative electrical synapse as shown on the top. (**D**) Dendritic distances from the putative electrical synapse to the soma of Neuron 1 (Soma 1) are plotted against the distance to the soma of Neuron 2 (Soma 2) for each putative electrical synapse formed between two LC neurons. Correlation coefficient $r=0.79$, $p<0.01$. The slope of the straight line is 0.66. (**E**) The putative synapse number was plotted against the coupling coefficient for each pair. $r=0.86$, $p<0.05$. (**F**) The distances from each putative synapse to two somata were averaged and then plotted against the coupling coefficient of each pair. The coupling coefficient was normalized if a electrically coupled pair had multiple putative synapses. $r=-0.19$. (**G**) Comparing the dendritic distance of putative synapses to the soma and the putative synapse number between two homotypic electrical coupling types (FF-FF vs. MP-MP: $p=0.19$ in the dendritic distance; $p=0.43$ in the putative synapse number).

putative synapses formed between the dendrite and the soma (*Figure 5B*). No putative synapses forming between two somata were identified in our data. Put together, our data suggest there was no preferential synaptic location for either FF–FF coupled pairs or MP-MP coupled pairs.

The dendritic branches where the putative synapse was localized were generally first or second-order branches and the putative synapses were most frequently located in the middle of the entire

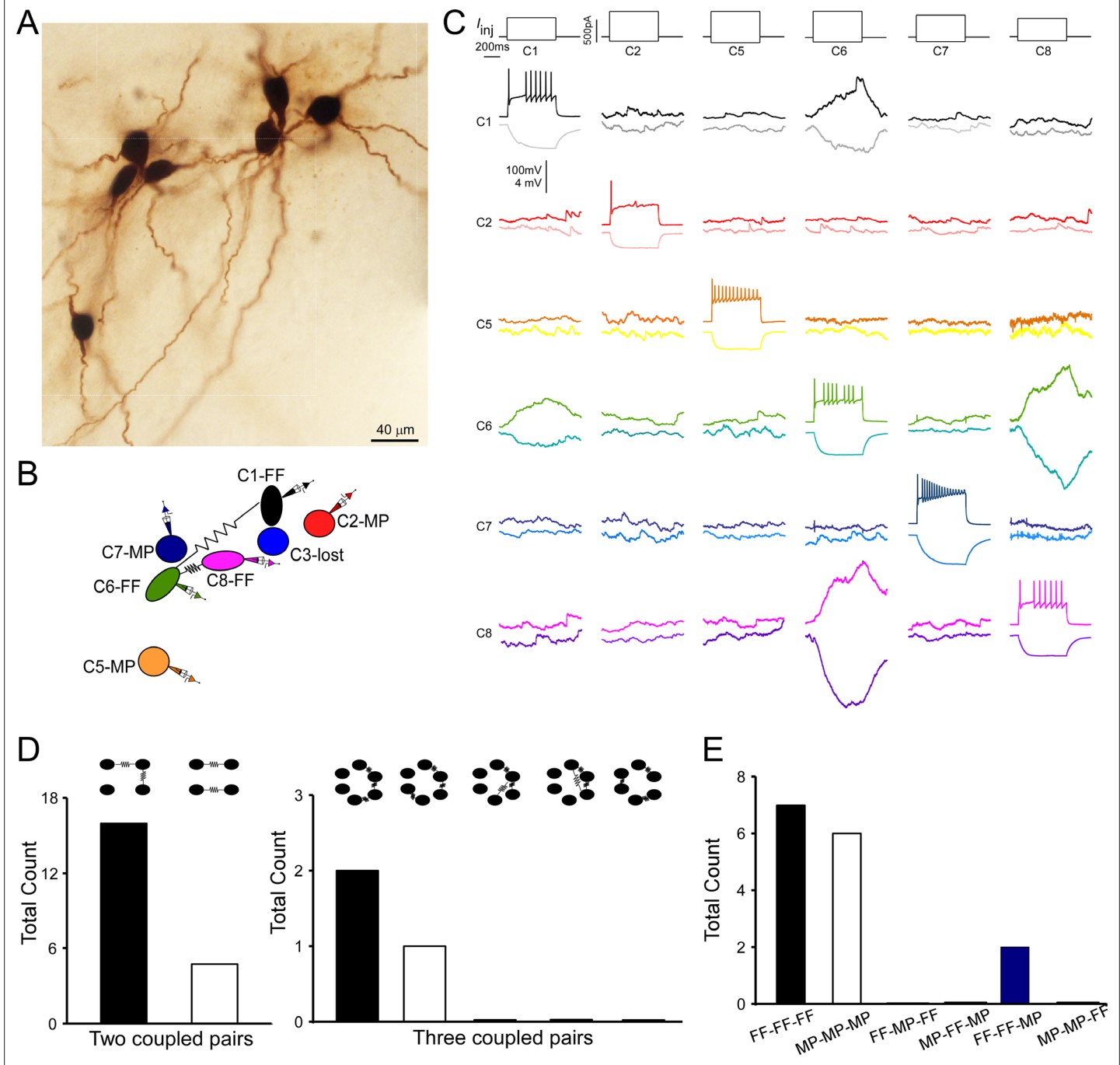

**Figure 6.** Multiple electrically coupled pairs identified in a cohort of locus coeruleus/norepinephrine (LC/NE) neurons. (**A**) A micrograph shows six morphologically recovered LC/NE neurons. (**B**) The relative position of six neurons (with assigned number) depicted as above and their connections. (**C**) Pairwise testing between six cells as shown in (**A**) reveals electrical coupling between Cell 1 and Cell 6, and between Cell 6 and Cell 8. No sign of electrical coupling between the rest of cell pairs. Vertical scale bar, 4 mV for the potential traces with no current injections, 100 mV for the traces evoked by injected currents. (**D**) The counts of multiple electrically coupled pairs identified from individual multicell recording sets with all possible arranging patterns (top). (**E**) The cell-type identity of each cell in those inter-connected coupled pairs presented in (**D**), and the frequency for each possible combination.

dendritic tree (average dendritic distance to soma: 155 μm, *Figure 5G*). When one cell pair had multiple contacts, these contacts were generally confined to the same major dendrite from each cell; some of them were even confined to the same dendritic branch of the same order of each cell. For each putative synapse, the dendric distance to one soma was linearly related to the distance to

another soma with an r of 0.86, indicating that the putative synapses sit a similar dendritic distance away from the two somata of the coupled pair (*Figure 5D*). The average dendritic distance of putative synapses from FF–FF pairs was 168 µm, tending to be slightly greater than the distance for MP–MP pairs (140.0 µm, *Figure 5G*). In addition, the dendritic distance of the putative synapse appeared to negatively correlate with its coupling coefficient (*Figure 5F*).

## Connectivity patterns of electrical coupling beyond two LC neurons

Multicell recordings (≥3 cells) allowed for examining electrical coupling beyond two cells. In many individual recording sets, we could identify multiple electrical connections among a cohort of cells. An example of such a case is illustrated in *Figure 6A–C*, in which a current pulse in presynaptic Cell 6 evoked a membrane response of the same sign in postsynaptic Cell 1 and 8. A current pulse in presynaptic Cell 1 or 8 evoked a membrane response of the same sign in postsynaptic Cell 6, but not 8 or 1 respectively. This indicates Cell 1 and Cell 8 both electrically connect with Cell 6, but Cell 1 and Cell 8 do not directly connect with each other. This result also indicates that electrical signals from one cell hardly pass to other cells via second-order coupling if they are not electrically coupled in the first order. Other tested pairs did not show evidence of electrical coupling in either direction (*Figure 6C*).

Out of our all multiple-cell recording sets (≥3 cells), 15 individual recording sets have more than two electrical connections (or electrically coupled pairs) identified. In these recording sets, one coupled pair often shared the same coupling partner with another coupled pair as illustrated in *Figure 6A–C* (i.e., A↔B↔C; each letter represents a cell; *Figure 6D*). This is surprising since two coupled pairs should be less likely to share the same partner if each cell is coupled with other cells in the network at the same rate. However, two coupled pairs without a sharing partner (i.e., A↔B, C↔D) were rare in our recording sets (*Figure 6D*). These data suggest each cell is not equally coupled with other cells in the network; instead, electrically coupled cells tend to cluster together and form a chain-like pattern (i.e., A↔B↔C). To test this hypothesis, we started by assessing how well a randomly connected network describes our dataset (*Erdos and Rényi, 1984*). In this model, the existence of an electrical connection between any two neurons was independently chosen with a uniform probability p (i.e., the overall connectivity rate across all of our recordings: 0.0438; *Figure 3D*). We first tested this random connectivity model in predicting the patterns of two electrical connections in our recording sets (Model 1, see 'Materials and methods'). Specifically, based on random pairwise connectivity, we predicted the numbers of a chain-like pattern of two connections (A↔B↔C) versus a non-chain-like pattern of two connections (A↔B, C↔D) using our recording sets of 3–8 neurons. We found that the observed number of the chain-like connection pattern (n=16, *Figure 6D*) was higher than the numbers predicted by the random connectivity model (n=12.9, 95% confidence interval [CI] = [6, 21]), whereas the observed number of the non-chain-like patterns (n=5, *Figure 6D*) was significantly less than the numbers predicted by the model (n=11.6, 95% CI = [6, 19], p=0.01). We calculated the ratio of the chain-like pattern number to the non-chain-like pattern number to reflect the clustering degree of two electrical connections, which we henceforth called the cluster index (*Figure 7A*). The cluster index in our experimental dataset was 3.2, which was significantly higher than the number predicted by the model (1.1, p=0.001). We then tested whether differences in the organizing pattern of two connections (observed vs. predicted) can be better explained by the cell-type-specific connectivity rule as we deciphered (see *Figure 4G*) by incorporating cell-type information into the model. That is to say, pairwise connection probabilities for FF–FF, FF–MP, and MP–MP pairs (see *Figure 4G*) were used to obtain a new model prediction. With this revised model (Model 2, see 'Materials and methods'), the observed numbers on average of the chain-like pattern again remained higher than the predicted numbers (n=12.6, 95% CI = [7, 18]), and the observed number of the non-chain-like pattern was still significantly lower than the numbers (n=11.2, 95% CI = [6,16], p<0.001) predicted by the model (*Figure 7B*). The cluster index predicted by Model 2 was very similar to that predicted by Model 1, and significantly lower than the observed index (*Figure 7A*), indicating that the cell-type-specific rule cannot explain our dataset. Given that neurons of the same type tended to be near each other (*Figure 2—figure supplement 4*) and electrical connections are highly sensitive to the inter-soma distance (see *Figure 4H*), we further revised our model (Model 3, see 'Materials and methods') by zeroing out connections between cell pairs that were greater than 80 µm, and by using the FF–FF and MP–MP connection rate for cell pairs within 40–80 µm apart and FF–MP connection rate for cell pairs within 80 µm apart. The results did not fundamentally change: the observed number of the

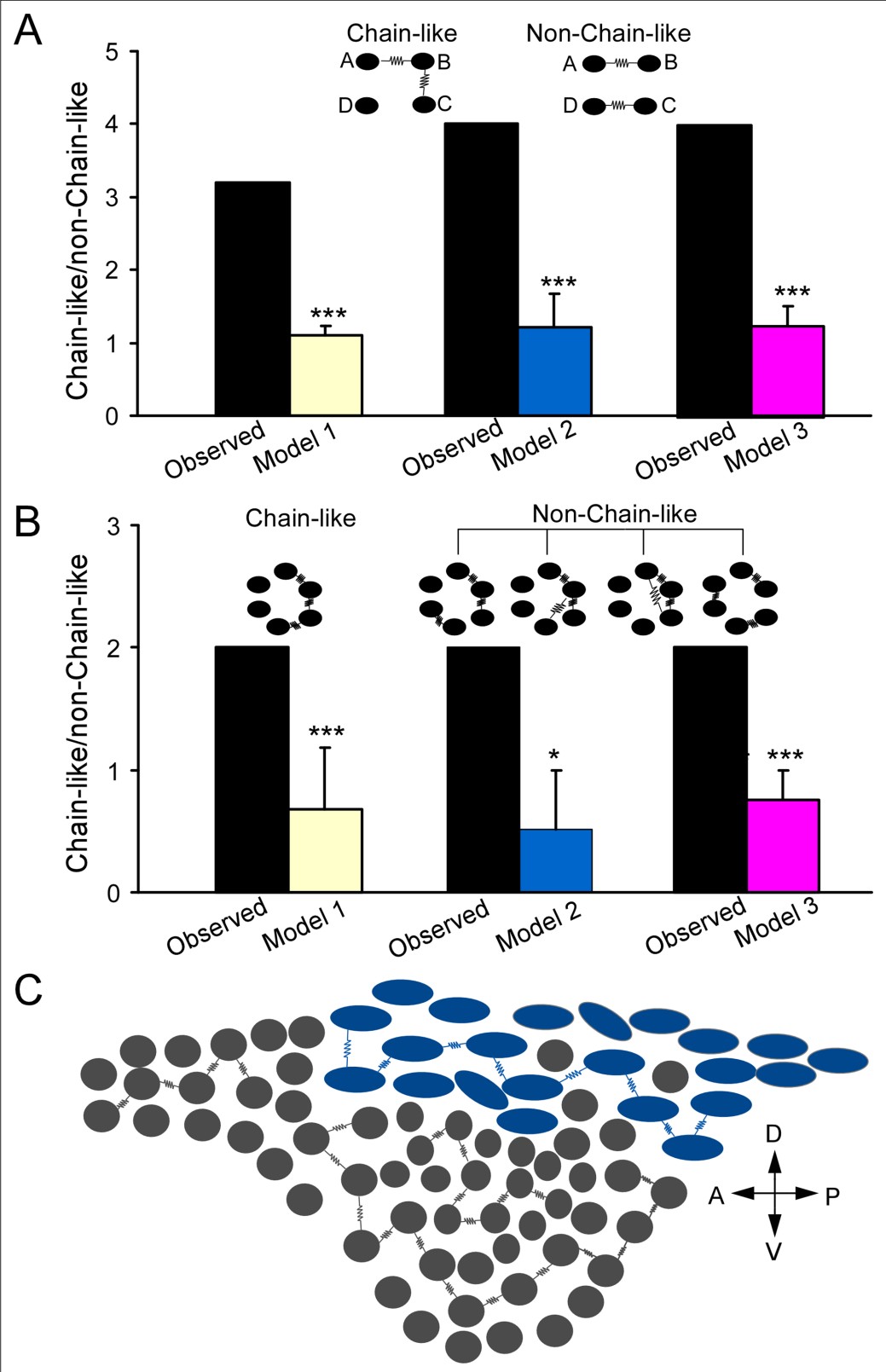

**Figure 7.** Electrically coupled homotypic locus coeruleus (LC) subnetworks. (**A**) In two electrical connections between LC neurons, comparing observed ratio (chain/non-chain pattern) vs the ratios predicted by Model 1 (random connectivity), Model 2 (cell-type-specific), and Model 3 (cell type and spatial-specific) suggests a chain-like rule dictating electrical connections. (**B**) In three electrical connections between LC neurons, observed ratio (chain/

*Figure 7 continued on next page*

*Figure 7 continued*

non-chain pattern) vs. the ratios predicted by Model 1 (random connectivity), Model 2 (cell-type-specific), and Model 3 suggests a chain-like rule dictating electrical connections. Error bars are standard deviations estimated by bootstrap method. ***p≤0.001, *p<0.05. (**C**) A sagittal view of the LC with predominant cell types for each subdivision (adopted from *Loughlin et al., 1986*) and the proposed electrically coupled homotypic subnetworks in the LC. Spindle-shaped: fusiform cell; round-shaped: multipolar cells. Each electrical connection form between cell pairs of same type and each connected pair is connected with each other like a chain within each LC subregion. A: anterior; P: posterior; V: ventral; D: dorsal.

chain-like pattern was not significantly different from the predicted value (n=19.4), but the observed number of the non-chain-like pattern (n=4) was still significantly lower than that (n=15.9) predicted by the model. The cluster index predicted by Model 3 was still significantly lower than the observed index (*Figure 7A*), indicating that additional spatial connectivity rules could not explain our dataset as well. In summary, the clustering of two connections in our dataset (i.e., a chain-like pattern) cannot be explained by the random model and spatial and cell-type-specific rules as shown above, suggesting an additional, possible chain-like rule dictating the organization of electrical connections in the LC network (note that all three models shared the assumption that the probability for two neurons to form an electrical connection is independent of all other pairwise connections, an assumption that the new rule will perhaps have to violate). Indeed, in a proof-of-concept new model (Model 4, Appendix 1) where the said assumption in Models 1–3 was violated (i.e., a cell's connectivity being scaled up or down contingent on whether or not it has an existing electrical connection), the observed patterns of two connections can be well predicted by the model (Appendix 1), supporting that electrical connections among LC neurons are not random, but incline to cluster together in LC (in addition to following cell type and spatial rules). Such an underlying new rule could impose on both cell types, and thus the most frequent patterns of two electrical connections in our dataset are FF–FF–FF and MP–MP–MP, while other combination patterns are either rare or not observed (*Figure 6E*).

Frequent observation of the clustered, chain-like organization of two connections suggests that an electrically coupled neuronal network may be wired as a chain-like network structure (*Figure 7C*). Indeed, when we examined the organizing pattern of three electrical connections in those individual recording sets, the most frequent was still the chain-like pattern (A↔B↔C↔D, *Figure 6D*), while other possible organizing patterns including circular-like and hub cell-like patterns were rarely observed in our dataset (*Figure 6D*). We then extended our models to predict how three electrical connections are organized. Our random connectivity model (Model 1) predicted 2.0 chain-like patterns of triple connections on average (95% CI = [0,6]) across all our recordings, which was the same as the actual observed number, while it predicted 3.0 non-chain-like triple connections on average (95% CI = [0, 7]), which was greater than the observed number (1). We again used the ratio of chain-like to non-chain-like patterns to measure the chain-like tendency in our dataset and compare it with model predictions. The observed ratio in our dataset was 2, significantly higher than the predicted ratio (0.68±0.25, p=0.001, *Figure 7B*). Our Model 2 (see above) also predicted 2.2 chain-like pattern of triple connections on average (95% CI = [0,6]); however, it predicted 4.9 non-chain-like triple connections (95% CI = [2,8]), which was significantly greater than the observed number (n=1, p=0.01). Again, the observed ratio was significantly higher than the ratio predicted by Model 2 (0.52±0.47, p<0.05, *Figure 7B*). Similar results were obtained using our Model 3 (see above, incorporating inter-neuronal distance), which predicted 6.8 of the chain-like patterns of triple connections on average and 9.3 non-chain-like triple connections on average. Once again, the observed ratio was significantly higher than the ratio predicted by the third and final model (0.76±0.24, p<0.001, *Figure 7B*). All simulations indicate that, in addition to spatial and cell-type-specific rule, an additional chain-like rule may apply for the organization of three electrical connections.

## Discussion

In this study, we overcame the technical barriers to record up to eight noradrenergic neurons simultaneously from adult mouse LC. Simultaneous multicell patch-clamp recordings (i.e., recording more than two cells) have been developed to study cortical circuits for a while, but it remains a challenge to apply this technique to small nuclei in the brainstem such as the LC given the space restraint and

limited cell number in each brain slice. By optimizing slice quality and adapting our multipatching system to small brainstem slices, we could simultaneously record LC/NE neurons in vitro in the cohort of an average of six cells from adult mice. This technical development allowed for profiling the cellular diversity of LC/NE neurons while simultaneously mapping their connections at a single-cell resolution. In fact, this study may represent the first multicell patch recordings of a monoamine modulatory system in the brain, allowing for examining how monoaminergic neurons interact with each other at the single-cell level for the first time. Our results indicate noradrenergic neurons in the LC could be classified into at least two major morpho-electric cell types, and neurons within each cell type form a homotypic network module via gap junctions. Given that each of these cell types may have preferential anatomical locations in LC and different projection targets, each electrically coupled homotypic network may coordinate/synergize their input/output as a whole to engage in distinct functions of the circuits within its efferent domain. In addition, our large dataset has allowed us to directly test several long-standing questions regarding local circuit interaction in the LC, including α2AR-mediated actions and fast glutaminergic synaptic connection at the single-cell level.

## Morpho-electric diversity of LC/NE neurons

Morphological heterogeneity of LC neurons has been noticed since the very early LC studies. The anterior horn of the LC and the ventral part of the LC core appears to be dominated by cells with a stellate multipolar soma, whereas its posterior horn and the dorsal edge of the LC core are dominated by cells with a fusiform-shaped soma (*Loughlin et al., 1986*; *Swanson, 1976*). In addition, LC neurons with distinct soma shapes preferentially project to different brain areas (*Loughlin et al., 1986*; *Waterhouse et al., 1983*), warranting further exploration of the morphological diversity of LC neurons and their functional relevance. To this end, we recovered the morphology of a large number of LC/NE neurons via biocytin staining to characterize their morphological features, providing a quantitative, less ambiguous classification of LC/NE neurons. We confirmed that the soma shape of LC/NC neurons could be either fusiform, round, or pyramid (*Loughlin et al., 1986*; *Swanson, 1976*). Cells with fusiform-shaped soma have a polarized dendritic arborization (bitufted or bipolar), while cells with round or pyramid-like soma have much less polarized dendrite arborization. These differences allowed for classifying LC neurons into two morphological cell types – multipolar cell (MP) and fusiform cell (FF), which was substantiated by a machine learning algorithm. In addition to having distinct somatodendritic shapes, MP and FF cells have different local axonal arborization as well, further supporting our classification. Finally, MPs and FFs have distinct intrinsic electrophysiological properties. While mature LC/NE neurons as a whole exhibit the stereotypical physiological properties that clearly differentiate them from hippocampal neurons, there are subtle differences between MP and FF in their individual APs: MPs had a longer AP with smaller amplitude than FFs. Furthermore, FFs could sustain higher frequency firing than MPs with large suprathreshold depolarizing current injections.

The observation that FFs fire narrower spikes than MPs suggests FF may correspond to 'narrow' units identified during extracellular recordings of LC neurons, while MP may correspond to those 'wide' units (*Su and Cohen, 2022*; *Totah et al., 2018*). However, 'narrow' units are heavily biased toward the ventral part of the LC (*Su and Cohen, 2022*; *Totah et al., 2018*), while FFs as we described here may be dorsally biased (*Loughlin et al., 1986*). In addition, FFs do not exhibit a higher spontaneous firing rate than MPs (*Figure 2H*), inconsistent with a higher firing rate found for extracellularly recorded 'narrow' units (*Totah et al., 2018*). Furthermore, there is a twofold difference in spike width between 'narrow' and 'wide' units (*Totah et al., 2018*), but the difference in spike width between FF and MP is subtle despite being significant (*Li et al., 2016*). Therefore, there is no consistent evidence supporting that FF cells we defined here correspond to those 'narrow' units, and MPs correspond to 'wild' units. Likely, each single unit type is not confined to a specific morphological type, or both single unit types may be from the same MP type reflecting within-MP functional heterogeneity. The latter may be more likely given that MPs are a predominant LC cell type and are more heterogeneous than FF in terms of soma size and spatial distribution (*Loughlin et al., 1986*), as well as electrophysiology properties including AP half-width (data not shown). Nevertheless, how in vitro intracellular waveforms correlate with in vivo extracellular ones is unknown in LC, and we thus cannot rule out the possibility that FF cells correspond to those 'narrow' units. Establishing a real correspondence between cell types as we discovered here and single unit types is important for the field, warranting a new systematic study (*Su and Cohen, 2022*; *Totah et al., 2018*; *Totah et al., 2019*).

Each of the two cell types has unique morphology and electrophysiology, suggesting that they may participate in distinct functions of the circuits within their efferent domains. Indeed, previous studies indicate that fusiform-shaped cells in the posterior horn of the LC and the dorsal edge of the LC core preferentially project to the neocortex and hippocampus, while multipolar neurons in its anterior horn and/or the ventral part of the LC core preferentially project to the spinal cord, cerebellum, and hypothalamus (*Loughlin et al., 1986*; *Waterhouse et al., 1983*). While it is difficult to register each cell in our slice recordings to their anatomical location in vivo, we reason that FF cells as we defined here correspond to those fusiform-shaped cells in the posterior horn of the LC and/or the dorsal edge of the LC core, while MP cells may correspond to those multipolar cells in the anterior horn and/or the ventral part of the core. If this is true, these two types as we defined here might form separate LC subnetworks with distinct projection domains, each performing a nonoverlapping function in the brain. In addition, FF and MP may be further classified into different subtypes based on the locations, each of which has different preferred projection targets as well and thus form distinct MP and FF subnetworks (*Loughlin et al., 1986*). This may be particularly the case for MP (also see below). To explore the functional relevance of these two major morphological types or the putative subtypes within each morphological type, identifying a specific genetic marker for each cell type or subtype is required. Prior studies indicate some LC neurons could express neuropeptide Y (NPY) and/or galanin. However, it appears neither NPY-expressing nor galanin-expressing LC neurons are confined to a specific morphological type (*Holets et al., 1988*; *Melander et al., 1986*), suggesting that neither NPY nor galanin is a good candidate to differentiate the two LC morphological types. In this regard, relatively new approaches such as single-cell RNA-sequencing (scRNA-seq) are being applied to the LC, which has the potential to identify the specific marker genes for each LC type or subtype (*Luskin et al., 2022*; *Scala et al., 2021*; *Tasic et al., 2016*; *Zeisel et al., 2015*). With type-specific marker genes identified with scRNA-seq, one can leverage an intersectional genetic strategy employing a dual recombinase system using both Cre recombinase as well as a second recombinase, flippase (Flp), to express the reporter genes in each specific LC neuronal type (or subtype). For this purpose, triple transgenic mice could be generated by crossing the Cre/Flp or Cre/Dre double-dependent reporters crossed with *Dbh*-cre (labeling all noradrenergic neurons) and an Flp driver line (expressing Flp in the marker gene locus) to label and manipulate each cell type (or subtype) while animals are trained to perform behavioral tasks (*Lusk et al., 2022*; *Madisen et al., 2015*; *Plummer et al., 2015*).

## Chemical transmission between LC/NE neurons

The physiological function of monoamine neurons is considered to be modulatory given that most monoamine axon terminals in the CNS are 'asynaptic' free nerve endings thought to mediate 'volume transmission.' However, increasing evidence suggest monoamine axon terminals, including noradrenergic terminals, co-release glutamate to participate in fast synaptic transmission. At least a subset of their axonal terminals makes the junctional or conventional-looking synaptic contacts onto targeted neurons, possibly mediating rapid glutamatergic synaptic transmission (*Fung et al., 1994a*; *Liu et al., 1995*; *Trudeau, 2004*; *Yang et al., 2021*). While evidence for this emerging concept of monoamine function is pretty strong (*Yang et al., 2021*), a direct demonstration of the glutamatergic nature of synapses established by monoamine neurons is still lacking. We thus took advantage of our unprecedented multicell recordings of monoamine neurons to examine whether there is any glutamatergic synaptic transmission between LC/NE neurons. Given the high-throughput nature of our system, we were able to test more than 1500 cell pairs, including those narrowly spaced cell pairs (<40 μm apart). However, we did not find evidence of glutamatergic synaptic transmission between LC/NE neurons in our horizontal slice preparation. It will be interesting to examine whether this type of connection, if it does exist among LC neurons, can be more readily detected in coronal slices where peri-LC glutamate transmission may be somewhat more spared than in horizontal slices. Nevertheless, we believe this type of connection, if it does exist, is sparse, given that only a small subset of all terminals established by monoamine neurons may actually be specialized for the synaptic release of glutamate (*Trudeau, 2004*). In addition, our result does not rule out the possibility that a subset of noradrenergic axonal terminals does make glutamatergic synapses onto other target regions, including the spinal cord and the parabrachial nucleus neurons (*Fung et al., 1994a*; *Liu et al., 1995*; *Yang et al., 2021*).

Monoamine release is controlled by their autoreceptors in the presynaptic terminals. The same receptors are often expressed in the somatodendritic domains of monoamine neurons, activation of

which increases potassium conductance as seen in their terminals (*Courtney and Ford, 2014*). Alpha2 adrenoceptors (α2AR) are such autoreceptors for NE which are expressed in the somata of LC/NE neurons as well. While α2AR-mediated action on LC neurons is well established, it is unclear under what conditions these receptors are activated and what is their natural agonist. These receptors may be expressed in the soma to sense NE locally released from other or the same LC/NE neurons to engage in lateral inhibition or autoinhibition, and/or epinephrine released from the medulla oblongata (*Cedarbaum and Aghajanian, 1976*; *Höˋkfelt et al., 1974*). It appears that NE could be released locally from local axonal terminals or somatodendritic domains of LC neurons following a single AP, but it is unknown whether NE released from individual neurons is sufficient enough to activate α2AR to evoke lateral inhibition on adjacent neurons. After testing more than 1500 cell pairs, we did not find any detectable hyperpolarization following the stimulation of a single LC/NE neuron. While other unknown reasons may contribute to the absence of lateral inhibition, a plausible explanation is that NE released from individual LC/NE neurons is not sufficient to activate α2AR to evoke detectable inhibition on LC/NE neurons at least in our in vitro preparation. A recent study suggests that α2AR-mediated lateral inhibition is a frequency-dependent population-wide phenomenon, which can be readily detected when many neurons are simultaneously activated via an optogenetic approach (*Davy et al., 2022*). In addition, α2AR-mediated lateral inhibition appears to be a specific network interaction that is prevalent among LC neurons targeting distinct brain regions (module-specific), rather than a nonspecific, spatial 'surround' inhibition (*Davy et al., 2022*). All these could explain why lateral inhibition could not be readily detected at the single-cell level in our dataset where each LC neuron was randomly targeted and individually tested for this type of interaction.

## Electrical connection between LC/NE neurons

Several previous studies suggest that electrical coupling in the LC is negligible or absent in adult animals (*Christie and Jelinek, 1993*; *Christie et al., 1989*; *Travagli et al., 1995*). This is surprising given that electrical coupling is an appealing mechanism for LC/NE neurons to synchronize to cope with the global demand for NE signaling. In contrast to these studies, our multicell recordings provide a direct demonstration of electrical coupling established by mature LC/NE neurons with a coupling rate consistent with a previous report from extracellular recording (*Totah et al., 2018*), substantiating prior evidence that electrical coupling is still an important mechanism for LC/NE neurons to interact with each other in adult animals (*Alvarez et al., 2002*; *Rash et al., 2007*; *Van Bockstaele et al., 2004*). An electrical coupling rate ranging from 15% to 80% has been reported in many brain regions based on pairwise testing of narrowly spaced cells of the same cell type from juvenile animals (*Curti et al., 2012*; *Devor and Yarom, 2002*; *Long et al., 2004*; *Tamás et al., 2000*). These coupling rates as reported in diverse brain regions may significantly drop e as cells are more spatially spaced Indeed, several studies indicate the electrical coupling rate often drops to zero once the inter-soma distance increases to more than 40 µm (*Beierlein et al., 2000*; *Curti et al., 2012*; *Galarreta and Hestrin, 2001*; *Galarreta and Hestrin, 2002*; *Gibson et al., 1999*; *Tamás et al., 2000*). These spatial and cell-type-specific rules appear to apply to electrical coupling among mature LC neurons as well. While the overall coupling rate is low among all LC cell pairs, it is much higher when cell pairs are narrowly spaced (with inter-soma distance <40 µm), particularly for FF-FFs. Interestingly, when a cell pair is more than 40 µm apart, the possibility of finding an electrical connection, despite being very low, is much less distance-dependent, possibly due to the dominant dendrodendritic form of electrical synapse formed between these neurons (see below). In addition, electrical coupling appears to decrease with age in most brain structures (*Peinado et al., 1993*), including in LC (*Christie and Jelinek, 1993*; *Christie et al., 1989*; *Travagli et al., 1995*). However, a direct comparison of the coupling rate in adults to juvenile animals did not reveal a significant difference between the two ages, indicating that electrical coupling in the LC may be less subject to developmental change than previously thought.

It is well documented that gap junctions generally form between cell pairs of the same type (homotypic gap junctions), while gap junctions between different cell types (heterotypic gap junctions) are rare or absent in most brain regions (*Peinado et al., 1993*). This rule also applies to LC neurons: most electrical couplings occur between cell pairs of the same noradrenergic type, while electrical coupling is rare between cell pairs of the different cell types. This is still true when the inter-soma distance is controlled. Comparing the spatial profiles of homotypic gap junctions versus heterotypic gap junctions, we found that neurons appear to form gap junctions exclusively with narrowly spaced, adjacent

LC neurons of the same type (homotypic connection), while the heterotypic connections, despite being very rare, typically form between cell pairs that are more spatially apart. The result suggests that each LC neuron is intricately connected with their very close neighboring neurons of the same type to constitute cell-type-specific electrical subnetworks (FF subnetwork vs. MP subnetwork), each of which may synchronize independently via extensive gap junctions. This finding provides a plausible cellular mechanism for prior observations that LC pairwise correlated firing and LC ensembles identified in extracellular recordings are cell-type specific (*Noei et al., 2022*; *Totah et al., 2018*). Of note, the coupling rate between FF–FF pairs is more than threefold higher than that between MP-MP pairs, suggesting that the MP electrical subnetwork may be smaller in terms of cell population than the FF subnetwork. Alternatively, a lower coupling rate among MPs may suggest that MPs are more heterogeneous than FFs. Indeed, as discussed above, LC neurons with multipolar-shaped soma are widely distributed across multiple LC subregions and more heterogeneous than those fusiform cells. In addition, putative MPs in distinct LC subregions appear to have different preferred projection targets (*Loughlin et al., 1986*; *Waterhouse et al., 1983*). Therefore, MP cells in each LC subregion may only electrically connect with MP cells in the same LC subregion to form subregion-specific MP subnetworks, each of whicht performs unique functions (*Figure 7C*). Given that FFs and MPs (or their subgroups) may have different preferential locations and distinct projection targets, within-cell-type electrical coupling is likely to promote the spread of electrical and chemical signals and spike synchronization within each subnetwork, exerting their synergetic effect on specific target regions.

While close appositions between two neurons do not necessarily mean electrical synapses, such contact sites identified under the light microscope could be reliably confirmed as real electrical synapses by follow-up electron microscope studies (*Fukuda and Kosaka, 2000*; *Fukuda and Kosaka, 2003*; *Tamás et al., 2000*; *Vervaeke et al., 2012*). Our examination of close appositions between electrically coupled pairs indicates more than 50% of electrically coupled pairs have only one putative synapse, and multiple contact sites between pairs are less frequent. Almost all putative synapses were found to occur between dendrites, corroborating a previous LC study in rats (*Ishimatsu and Williams, 1996*). In fact, dendro-dendritic synapses appear to be a dominant form of electrical synapses across brain regions (*Fukuda and Kosaka, 2000*; *Fukuda and Kosaka, 2003*; *Hidaka et al., 2004*; *Sotelo et al., 1974*; *Vervaeke et al., 2010*; *Waterhouse et al., 1983*), while somato-dendritic or somato-somatic ones are much less common (*Curti et al., 2012*; *Tamás et al., 2000*). In addition, electrical synapses in dendrites are localized further away from the soma as animals get older (*Fukuda and Kosaka, 2000*; *Fukuda and Kosaka, 2003*). Consistent with these previous findings, putative synapses in mature LC were located most frequently in the middle part of the entire dendritic trees of the electrically coupled cell pair, with a similar dendritic distance to each soma. Given electrical synapses being located at such distal sites, fast signals like action potentials in prejunctional cells could be low-pass filtered and highly attenuated when propagating to postjunctional cells, giving rise to small-amplitude coupled potentials (like spikelet) at the somata of post-junctional neurons. How could such a small coupled potential drive synchronized firing and network oscillations under natural stimulation conditions? Recent work has shown those gap junctions found at distal dendritic sites can interact locally with chemical synaptic inputs to drive synchronized spiking. While coupled potentials are small at the soma, these electrical signals are expected to be larger near gap junctions, where they enable excitatory synaptic charge to spread into the dendrites of neighboring coupled neurons (*Vervaeke et al., 2012*), and/or sum with chemical synaptic inputs to generate dendritic spikes that actively propagate with little delay to the soma and drive action potentials with high fidelity. The former may promote the synchronized response of an electrically coupled neuronal population to certain specific synaptic inputs, while the latter allows weak distal gap junction inputs to synchronize somatic action potentials, promoting synchronized spiking of electrically coupled neurons upon common chemical synaptic input (*Migliore et al., 2005*; *Schoppa and Westbrook, 2002*; *Trenholm et al., 2014*). It will be interesting to investigate in future studies the mechanism(s) that LC neurons deploy to leverage their gap junctions to promote inter-neuronal synchrony.

## The network of electrically coupled LC/NE neurons

Electrical coupling is not just about connections between two neurons, instead it connects a group of neurons into an electrically coupled network that coordinates/synergizes their activity as a whole. Thus, two questions ensue: what is the population size of each network, and how does an individual neuron

in each network connect to other neurons of the same network? Despite extensive studies of gap junctions across almost all brain regions, these questions remain largely unexplored due to technical limitations. Gap junction studies with the dye-coupling approach may provide an insight into these questions (*Curti et al., 2012*; *Long et al., 2004*), but the neuron clusters labeled by this approach could result from a second-order coupling, in addition to first-order coupling – these couplings are not otherwise discriminable – greatly compounding interpretation. In addition, the relative impermeability of different dyes to different kinds of electrical synapses results in many inconsistent results. In this regard, our multipatch recordings confer some technical advantages over the dye-based approach. First, given gap junctions being located at the dendrite far away from the soma, somatic membrane potentials in a pre-junctional cell are likely to be highly attenuated by the time they reach a post-junctional cell. It is reasonable to conclude that such highly attenuated coupled potential from the first-order coupling can hardly propagate any further into cells through second-order coupling. Thus, the gap junctions with a dominant dendrodendritic form identified with pairwise recordings should be essentially first-order gap junctions, but this assertion has never been directly confirmed as yet. Our multicell recording sets allowed for directly testing said assertion and ruled out the possibility that gap junctions identified with pairwise patch recordings in LC could result from a second-order coupling. However, caution should be taken in interpreting the results if electrical synapses are located in the soma (*Curti et al., 2012*), assuming the signals passing through somatosomatic electrical synapses are less attenuated. Given that the electrical couplings we identified in LC neurons are confirmed as first-order, we can then examine the real organization of electrical connections beyond two cells in the context of the network. Specifically, our multiple-cell recording allows for identifying multiple electrically coupled pairs across a cohort of up to eight cells, providing the first glimpse into how each neuron interacts with the rest of the neurons in an electrically coupled neuronal network.

With the support from the simulation with a random connectivity model, a chain-like pattern of two connections (A↔B↔C) is overrepresented in our dataset. One might argue that said overrepresentation may be accounted for by spatial and cell-type-specific connectivity rules that were empirically determined by us here. However, revising our model by adding these connectivity rules did not significantly improve simulation results, and a chain-like pattern of two connections (A↔B↔C) remains unexplained. All simulations suggest that, in addition to spatial and cell-type-specific rules governing LC electrical connections, there is an additional chain-like connectivity rule that may contribute to the overrepresentation of a chain-like pattern of two connections in our dataset. Furthermore, while the number of recording sets with three connections is low, a chain-like pattern (A↔B↔C↔D) is still a dominant pattern of three connections in our recording sets. Our simulation on the pattern of three connections yielded similar results and supported the idea of overrepresentation of a chain-like electrical connectivity pattern in LC/NE neurons. In addition, the overrepresentation can not be explained by the spatial and cell-type-specific connectivity rules, further suggesting an additional chain-like connectivity rule for the organization of three connections. While we could not test if this additional connectivity rule may apply to four or more connections for now (no four connections were found in our dataset), the chain-like connectivity rule may be a general wiring principle dictating a whole electrically coupled network composed of hundreds of neurons (*Figure 7C*). Indeed, previous dye-based studies suggest the same connectivity rule in many brain regions and cell types, including LC (*Christie and Jelinek, 1993*; *Curti et al., 2012*; *Lee et al., 2014*; *Long et al., 2004*). If a chain-like organizing pattern is true, then the dye injection of a single cell can only label the maximum of two additional cells if only first-order coupled cells are labeled. Indeed, the cluster of labeled cells in these studies is often small with a maximum of two additional labeled cells (*Christie and Jelinek, 1993*; *Curti et al., 2012*). Some studies do label more than two additional cells, but additional cells may be labeled via second-order coupling with a long dye injection time (*Devor and Yarom, 2002*; *Long et al., 2004*). The most convincing evidence for this rule comes from the mesencephalic trigeminal nucleus, where gap junctions between five neurons have been elegantly shown to arrange like a chain-like pattern across neurons (*Curti et al., 2012*). However, the interpretation of these data is restrained by several issues inherent to this approach (see above), and whether or not this chain-like connection pattern applies to the whole LC electrically coupled network needs to be further explored in the LC, as well as in many other brain regions. In addition, the other two possible EC organization patterns, circular (*Figure 6D*, right panel fourth cartoon) or hub-like connection (see *Figure 6D*, right panel third cartoon) patterns, were not observed in our dataset, suggesting that electrically coupled

pairs in LC may not be organized as these two patterns. However, simulated results on current data could not rule out these two patterns. Model 3 simulations predicted 2.45 hub-like connections on average (95% CI: [0,5]) and predicted 1.39 circular-like connections on average (95% CI: [0,4]). While both average predicted numbers are numerically higher than the observed numbers of each (0), the observed numbers are still within statistical bounds. More data collected with more advanced recording techniques may be able to answer these questions in the future.

## Materials and methods

### Animals

All experiments were performed according to the guidelines of the Institutional Animal Care and Use Committee (IACUC) of Baylor College of Medicine. Tg(Dbh-cre)KH212Gsat/Mmucd (henceforth *Dbh*-cre; C57BL/6 background) originally obtained from the Mutant Mouse Regional Resource Center at the University of California, Davis, CA (https://www.mmrrc.org/catalog/sds.php?mmrrc_id=36778). Crossing *Dbh*-Cre mice with Ai9 reporter mice (JAX stock no: 007909: B6.Cg-*Gt(ROSA)26Sor*$^{tm9(CAG-tdTomato)Hze}$/J) globally labels all noradrenergic neurons including LC neurons (*Rukhadze et al., 2017*). Experiments on adult male and female mice (median age: 79, full range: 55 to –175 days) were performed using *Dbh*-Cre: Ai9 (n=121) or wild-type (n=15). Additional younger *Dbh*-Cre: Ai9 mice (P13-21, n=13) were used to study connectivity between LC neurons at the juvenile age for comparison with the adult age.

### Slice preparation

Slice preparation from adult mouse brainstem follows an N-methyl-D-glucamine (NMDG) slicing protocol (*Jiang et al., 2015*; *Ting et al., 2014*). Briefly, animals were deeply anesthetized using 3% isoflurane. After decapitation, the brain was removed and placed into cold (0–4°C) oxygenated NMDG solution containing 93 mM NMDG, 93 mM HCl, 2.5 mM KCl, 1.2 mM NaH$_2$PO$_4$, 30 mM NaHCO$_3$, 20 mM HEPES, 25 mM glucose, 5 mM sodium ascorbate, 2 mM thiourea, 3 mM sodium pyruvate, 10 mM MgSO$_4$ and 0.5 mM CaCl$_2$, pH 7.35 (all from Sigma-Aldrich). Horizontal slices were prepared using a vibratome (200 μm thick) with zirconia ceramic blades. The brain slices were kept at 37.0 ± 0.5°C in oxygenated NMDG solution for 10 min, and then were transferred to an artificial cerebrospinal fluid (ACSF) containing 125 mM NaCl, 2.5 mM KCl, 1.25 mM NaH$_2$PO$_4$, 25 mM NaHCO$_3$, 1 mM MgCl$_2$, 25 mM glucose, and 2 mM CaCl$_2$ (pH 7.4) for at least 1 hr prior to the beginning of recordings. During the recording sessions, the slices were submerged in a custom chamber and were stabilized with a fine nylon net attached to a custom-designed platinum ring. This recording chamber was continuously perfused with oxygenated ACSF throughout the recording session. Brainstem slices from juvenile mice were prepared following a similar protocol except for the cutting solution (with regular ACSF, not NMDG). For the recording of hippocampal neurons, the parasagittal cortical slices were prepared following the same protocol for adult animals.

### Electrophysiology

Whole-cell recordings were performed as described previously (*Jiang et al., 2015*; *Scala et al., 2021*; *Scala et al., 2019*). Briefly, patch pipettes (4–7 MΩ) were filled with an internal solution containing 120 mM potassium gluconate, 10 mM HEPES, 4 mM KCl, 4 mM MgATP, 0.3 mM Na$_3$GTP, 10 mM sodium phosphocreatine, and 0.5% biocytin (pH 7.25). Simultaneous whole-cell recordings from up to eight neurons were performed using two Quadro EPC 10 amplifiers (HEKA Electronic, Germany). Patch-Master (HEKA) and custom-written MATLAB-based programs (MathWorks) were used to operate the recording system and perform online and offline data analysis. In current-clamp recordings, neurons were first current clamped at ~–40 pA to prevent its spontaneous firings (see 'Results'). Action potentials were evoked by current injection into presynaptic neurons at 2 nA for 2 ms at 0.05 Hz for 20–50 trials, and the average of the sweeps in postsynaptic cells was used to detect synaptic connections. The spike threshold and firing pattern in response to sustained depolarizing currents were recorded for each neuron by injecting increasing current steps (+10 pA).

For coupling coefficient (CC) calculations in gap junctions, hyperpolarizing current pulses of 600ms duration were injected into one cell and resulting voltage deflections were measured in both cells. The voltage deflections were measured at the last 10–20 ms of current injections given the large time

constant of the membrane potential responses of two cells (see 'Results'). The coupling coefficient from cell 1 to cell 2 (CC1) was defined as V2/V1 and the coupling coefficient in the opposite direction from cell 2 to cell 1 (CC2) was defined as V1/V2, where V1 is the voltage deflection in one cell and V2 is the corresponding voltage deflection in the other cell. To improve the signal-to-noise ratio, a total of 4–30 single responses were often generated and averaged. For each of the coupled pairs, the mean coupling coefficient was calculated as the average from the values in both directions.

## Electrophysiological features extraction

A custom-made Python software kit named Patchview was developed to analyze the electrophysiology data offline (*Ming Hu, 2022*). We extracted 13 electrophysiological properties from each neuron, following a similar method as used in previous studies (*Gouwens et al., 2019*; *Scala et al., 2021*). The rheobase refers to the minimal stimulation current to elicit any action potentials. To extract the properties of AP, we used the first AP elicited by our step stimulation protocol. To calculate the threshold of the AP, the trace within 20 ms preceding the AP peak was used to calculate the third derivative first and the data point with the maximal third derivative value was defined as the threshold of the AP. The AP amplitude is the difference between the threshold potential and the peak potential of the AP. The AP width was defined as that at half-height. The time range from the threshold point to the peak is 'rising phase time' while 'decay phase time' is the time range from the peak to the potential equal to the threshold. The afterhyperpolarization was defined as the voltage difference between the threshold and the minimal value of the trough after the AP. The time range from the stimulation onset to the threshold of the first AP at the rheobase sweep was defined as the AP delay time.

To calculate the time constant $\tau$, we fitted each hyperpolarizing trace (from the stimulation onset to the minimal potential within the first-half stimulation period) with an exponential function as follows:

$$f(t) = ae^{(-t/\tau)} + C$$

where $t$ is the time relative to the stimulation onset, $a$ and $C$ are two constants, $f(t)$ is the current value at time $t$, and $\tau$ is the membrane constant. Then the $\tau$ was calculated for each trace and averaged across all traces.

To calculate the maximal firing rate, the depolarizing trace with the maximal number of APs elicited by the step protocol was used to count the number of APs, which was then divided by the stimulation duration (600 ms). To calculate the adaptive index, the inter-spike intervals (ISIs) were first measured, and then the ratio of the last and the first ISIs was calculated. The CV was calculated as the ratio of the standard deviation of ISIs and the mean of ISIs. To compare the first AP and the subsequent ones, we calculated the ratio of their AP amplitude and the ratio of their half-widths. The amplitude and half-width of the first AP were first calculated and then divided by the mean of these values of the subsequent APs.

To calculate the input resistance, we measured the steady-state membrane potentials in the traces elicited by the first five negative currents in the step protocol, and then plotted these values (y-axis) against each of the corresponding currents (x-axis). We fitted the plot with a linear function and the slope of the fitting line is the input resistance, and the potential at the x-axis intersection is the RMP. The RMPs were reported without correcting junction potentials.

## Morphological reconstructions and analysis

Morphological reconstructions and analysis were performed using a ×100 oil immersion objective lens and camera lucida system using Neurolucida (MicroBrightField), following the same procedure as previously described (*Jiang et al., 2015*; *Scala et al., 2021*; *Scala et al., 2019*). Briefly, neurons were filled with biocytin during the recording, and slices were fixed right after recordings by immersion in freshly prepared 2.5% glutaraldehyde (from Electron Microscopy Science Cat# 16220), 4% paraformaldehyde (from Sigma-Aldrich Cat# P6148) in 0.1 M phosphate-buffered saline at 4°C for 7–10 days, and then processed with the avidin-biotin-peroxidase method to reveal cell morphology. Slices were then mounted in an aqueous Mowiol mounting solution (Sigma-Aldrich) and the morphologically recovered cells were examined, reconstructed, and analyzed using a ×100 oil-immersion objective lens and camera lucida system. Tissue shrinkage due to the fixation procedure was not compensated. The shrinkage of the tissue surrounding the biocytin-stained cells is about 10–20%, consistent with

previous studies (*Markram et al., 1997*; *Scala et al., 2019*). The reconstructions were not corrected for tissue shrinkage.

Dendritic arbor and axonal structure of reconstructed morphologies of n=70 cells were analyzed using Neurolucida software. Additional morphological analysis was performed using custom Python code.

## Data analysis

Data analysis was performed using custom-written MATLAB software and Python routines. For statistical comparisons, the normality of the data was first tested using the Shapiro–Wilk test. If compared groups show normal distributions, Student's *t*-test or the paired *t*-test was used; otherwise, the nonparametric paired Wilcoxon sign-rank test and Mann–Whitney test were used. One-way or two-way ANOVA was also used for comparing morpho-electric features of different types of neurons.

## Data simulation

### Model 1: Random connectivity model

First, the overall pairwise connection probability (P) was obtained from the recordings and is the ratio of the observed number of electrical connections to the total number of potential electrical connections tested across all recordings, regardless of the neuronal type and inter-neuronal distance. Expected numbers of double or two-connections were calculated based on pairwise connection probabilities. For instance, if there were four neurons A, B, C, and D in a particular recording/slice, then the expected value of a non-chain-like two-connection AB-CD was calculated to be equal to P(pairwise connection) * P(pairwise connection). The expected number of chain-like and non-chain-like two-connections for a given slice was the sum of the expected values of all two-connections for the recording. The grand total expected number of two-connections of a given type was the sum of the corresponding expected number values over all our recordings. Similar simulations were conducted for triple or three-connections, and the product of three pairwise connections was used for all calculations.

### Model 2: Differentiated neuronal types

For each individual recording (slice) for which the slice was fixed and neuronal types (FF – fusiform cell type; MP – multipolar cell type) were identified, we calculated the expected numbers of chain-like and non-chain-like two- and three-connections in the following way. For a given pair of neurons from a given recording slice, its corresponding pairwise connection probability (FF-FF, FF-MP, or MP-MP depending on the morphological cell types of the neurons constituting the pair) across all recordings, which was computed earlier, was used for all calculations (*Figure 4G*). Expected numbers of two-connections were calculated based on pairwise connection probabilities. For instance, if neurons A, B, C, and D in a particular recording/slice were FF, FF, MP, and MP neuronal types, respectively, then the expected value of a chain-like two-connection AB-BC was calculated to be equal to P(FF-FF connection) * P(FF-MP connection). Similar calculations were performed for all possible two-connections (e.g., chain-like two-connection AB-CD and non-chain-like connections AB-CD in the above example). The expected number of chain-like and non-chain-like two-connections for a given slice was then the sum of the individual probabilities thus calculated. The grand total expected number of two-connections of a given type was computed by summing the corresponding expected number values over all our recordings. Calculations to compute the expected values of triple or three-connections were similar, with the exception that the expected value was now a product of three, and not two, pairwise connections; in the above example, P(AB-BC-CD)=P(FF-FF connection) * P(FF-MP connection) * P (MP-MP connection).

### Model 3: Differentiated neuronal types, short-range connectivity

In addition to the constraints specified in Model 2, namely, pairwise connection probabilities based on neuronal identity, the following additional constraints were added to Model 3. First, because connections between neuronal pairs were largely nonexistent for inter-neuronal distances >80 μm, all neuronal pairs in the model that were >80 μm apart were taken to have no connection. Second, because neurons of the same type clustered together in the recordings, FF-FF and MP-MP connection probabilities for short distances (16–40 μm) were used, and because neurons of different types were

further apart in the recordings, FF-MP connection probability for 0–80 mm distance was used in the model. The remaining computations were performed in accord with those of the previous model.

## Monte Carlo simulations

On each of 1000 runs of the simulation, we counted the number of two- and three-connections from all simulated recordings. Each simulated recording matched one-to-one a biological recording in terms of the number and type of neurons. A random number generated by the random number generator determined whether a connection existed between each neuron in the corresponding simulated recording. If the random number was lower than the expected value of the connection (different values based on whether the corresponding connection in the recording was FF-FF/FF-MP/MP-MP), the simulated connection was considered to exist. For each run, simulated double and triple connections were obtained for each recording and the total simulated number was obtained by summing over all the simulated recordings. Summary statistics were obtained from the simulated data from 1000 such runs: (i) the mean, standard deviation, and 95% CIs for the simulated numbers of chain-like and non-chain-like two- and three-connections, and (ii) the probability that the ratio of the simulated chain-like over non-chain-like two-connections (three-connections) was greater than the ratio of the observed number of chain-like over non-chain-like two-connections (three-connections). All simulations were conducted on MATLAB (R2021a, The MathWorks, Natick, MA).

## Acknowledgements

We thank AM's thesis committee members for their suggestions and comments, as well as the comments and support from the members of Jiang Lab. We acknowledge funding from NINDS R01 NS101596 (XJ, AM, MH), NIMH R01 MH109556 (JJ, XJ, MH), as well as Training Grant T32 EY07001 (AM) to support this work. Research reported in this publication was also supported by the Eunice Kennedy Shriver National Institute of Child Health & Human Development of the National Institutes of Health under Award Number P50HD103555 for use of the Microscopy Core facilities. The content is solely the responsibility of the authors and does not necessarily represent the official views of the National Institutes of Health.

## Additional information

### Funding

| Funder | Grant reference number | Author |
| --- | --- | --- |
| Eunice Kennedy Shriver National Institute of Child Health and Human Development | P50HD103555 | Andrew McKinney |
| National Eye Institute | T32 EY07001 | Andrew McKinney |
| National Institute of Mental Health | MH109556 | Ming Hu Junzhan Jing Xiaolong Jiang |
| National Institute of Neurological Disorders and Stroke | NS101596 | Andrew McKinney Xiaolong Jiang |

The funders had no role in study design, data collection and interpretation, or the decision to submit the work for publication.

### Author contributions

Andrew McKinney, Data curation, Formal analysis, Investigation, Methodology; Ming Hu, Software, Formal analysis; Amber Hoskins, Arian Mohammadyar, Nabeeha Naeem, Saumil S Patel, Formal analysis; Junzhan Jing, Investigation; Bhavin R Sheth, Software, Formal analysis, Supervision, Investigation, Methodology; Xiaolong Jiang, Conceptualization, Resources, Data curation, Formal analysis,

Supervision, Funding acquisition, Validation, Investigation, Visualization, Writing - original draft, Project administration, Writing - review and editing

### Author ORCIDs
Junzhan Jing http://orcid.org/0000-0003-4647-0932
Xiaolong Jiang http://orcid.org/0000-0001-8066-1383

### Ethics
Animal housing and use were approved and in compliance with the guidelines of the Institutional Animal Care and Use Committee (IACUC) of Baylor College of Medicine.

### Decision letter and Author response
Decision letter https://doi.org/10.7554/eLife.80100.sa1
Author response https://doi.org/10.7554/eLife.80100.sa2

---

## Additional files

### Supplementary files
• MDAR checklist

### Data availability
All data generated or analyzed during this study are included in the manuscript and supporting files. The data and custom codes supporting the findings have been deposited in Dryad (https://doi.org/10.5061/dryad.kh1893283).

The following dataset was generated:

| Author(s) | Year | Dataset title | Dataset URL | Database and Identifier |
|---|---|---|---|---|
| Jiang X | 2023 | Cellular composition and circuit organization of mouse locus coeruleus | https://dx.doi.org/10.5061/dryad.kh1893283 | Dryad Digital Repository, 10.5061/dryad.kh1893283 |

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

## Appendix 1

### Model 4: Dependent pairwise connectivity model

Since simulations of all previous models with independent probabilities of pairwise connections failed to reproduce the experimental data (i.e., the numbers of the chain-like pattern of two connections were consistently underestimated by our models, whereas the numbers of the non-chain-like pattern of two connections were consistently overestimated), the above constraint was relaxed in a proof-of-concept new model. Here, the probability for a neuron to connect with another neuron was not independent of other connections in the recording. Instead of using an overall connectivity probability for all neurons (P: the ratio of the observed number of electrical connections to the total number of potential electrical connections tested across all recordings), we manipulated a neuron's connectivity as a function of other pairwise inter-neuronal connections in a given recording in two different ways. First, if a given neuron already had a pairwise electrical connection with another neuron, then the probability for it to form a pairwise electrical connection with yet another neuron would be scaled up by a factor $f_1$ (i.e., the new probability = $P*f_1$), whereas if the recording slice already had a pairwise connection between two neurons, none of which was the given neuron, then its probability of forming a pairwise connection would be scaled down by a factor $f_2$ (i.e., the new probability = $P/f_2$). The simulation was performed for every recording slice of n neurons (where n=2,3, …, or 8) with the total number of simulated slices equivalent to the number of slices in our recordings, so that the total number of possible pairwise connections across all the recording slices in our model is equal to the total number of said connections in the biology. Simulated numbers of double or two-connections were calculated as before. We found that for several different values of the scaling-up factor $f_1$ and scaling-down factor $f_2$, model estimates came close to experimental data. For instance, for $f_1$ and $f_2$ equal to 2.9 and 3.9, respectively, the observed number of the chain-like pattern (n=16, *Figure 6D*) was no different from the numbers predicted by the new dependent pairwise connectivity model (n=16.0, 95% CI = [6, 28]), and the observed number of the non-chain-like pattern (n=5, *Figure 6D*) was indistinguishable from the numbers predicted by the new model (n=4.8, 95% CI = [0, 12]). In further simulations, we found that one factor alone ($f_1$ or $f_2$) cannot reproduce the experimental findings. At present, we have no evidence for mechanisms that could potentially underlie either the scaling-up factor $f_1$ or the scaling-down factor $f_2$, but this simulation indicates that the probability for a neuron to connect with another neuron is contingent upon whether it has existing electrical connections, supporting that electrical connections among LC neurons are not random, but incline to cluster together in LC. We are now revising the new model by adding spatial and cell-type-specific rules into the model to test what $f_1$ and $f_2$ would be for the model to predict our experimental data.

