## [Editor Report]

Recent studies of the brainstem locus coeruleus (LC) noradrenaline system have demonstrated a modular functional organization, yet how different noradrenaline cell classes are independently regulated is not clear. Using ex-vivo, multi-patch recordings of up to eight LC neurons at once, this study offers compelling evidence for the existence of two morpho-electric cell classes with segregated electrical coupling. These important findings establish principles of local circuit communication occurring preferentially within defined LC-noradrenaline cell classes.

---

## [Decision Letter]

**Decision letter after peer review:**

Thank you for submitting your article "Cellular and Circuit Organization of the Locus Coeruleus of Adult Mice" for consideration by *eLife*. Your article has been reviewed by 2 peer reviewers, and the evaluation has been overseen by a Reviewing Editor and John Huguenard as the Senior Editor. The following individual involved in the review of your submission has agreed to reveal their identity: Nelson Totah (Reviewer #1).

Essential revisions:

1) Based on their findings, the authors argue that locus coeruleus noradrenaline neurons do not communicate through direct, excitatory synaptic transmission. However, this possibility has not been ruled out through their experiments as the horizontal slice preparation likely severed some axonal connections. The authors could try performing some experiments in coronal slices in which axonal connectivity may be somewhat more spared. Alternatively, they could moderate their claims in this aspect of the study.

2) The authors also argue that their results support the idea that lateral inhibition through alpha2 adrenergic receptors does not occur in the LC. However, there is a wealth of data supporting the existence of lateral inhibition and the present finding are based on the stimulation of single LC-noradrenaline neurons without a more parametric analysis of different stimulation frequencies/timing. For example, it is possible, given that LC neurons often fire together, that the activity of many LC neurons at one time is required to detect lateral inhibition, as noted by the authors in the discussion. In addition, given previous work showing that the frequency and timing of spikes are important in producing and detecting lateral inhibition (see Rev. 2 comments below and Huang et al. 2007. Proc National Acad Sci 104, 1401-1406), further exploration of this question is warranted. The authors could attempt stimulation of multiple LC-noradrenergic neurons at one time (possibly using optogenetic approaches) and/or explore the parametric space more extensively. Alternatively, they could remove this section from the paper.

*Reviewer #1 (Recommendations for the authors):*

Congratulations to the authors on an extremely difficult set of experiments, stunning anatomy, and careful analyses to address important and timely questions about the LC. Really (to my knowledge) no one has dug into the anatomy like this since a few Japanese groups in 1978/1979 and Americans in mid-1980's. An update was needed – and to couple them with recordings is a truly unique data set. I have some comments and questions which I hope will help the authors to improve the manuscript and connect it better with the existing literature.

One of the major strengths of the Author's work is the ability to potentially connect LC neuron types (FF and MP) with the two types of previously reported extracellular single unit. In light of that, additional characterization of a few things would be help to the field in making the connection to single units. The authors show clear data that FF cells may correspond to 'narrow units' because they have narrow AP width and have larger (albeit not significantly) AHP. Do FF have higher FR as do "narrow units"? Where were the FF neurons located in the LC core? The "narrow units" have been shown to be located throughout the core, but biased ventrally. This info would help link to existing extracellular study identifying multiple cell types.

It is a bit strange that Figure 1C and 1D come after Figure 2. It may improve the flow of the manuscript to move those panels into figure 2.

Figure 2E – it would be helpful to have a conceptual description of what it would mean to be low x value – low y value versus high x value – high y value.

Figure 2F – are the rings here the number of neurons? It would be helpful to the reader if the authors could label the axis and state this in the legend.

Figure 2H, 2I – it would help to explain that Y axis of panel is related to branching point. It is not clear to me how to interpret this plot. it might be standard in this field but would help the general reader to have this information briefly in the Results section or figure legend to better understand the plot.

Figure 3A ("Cell 7 and Cell 2 is electrically coupled") – "is" should be "are".

Figure 7C – This plot in C is confusing. The methods state that a horizontal (transverse) plane was used, therefore, ventral and dorsal should be medial and lateral. Can the Authors clarify this?

Line 78 – strike "the".

Line 94 – Can the authors comment on Why the RMP is so different from what was reported in earlier work (see Supplementary Table 5 in Chandler, Gao, and Waterhouse. 2014. Proc National Acad Sci 111, 6816-682)?

Lines 132-134 – The fact that the axon originated from different locations on FF versus MP cells is interesting. Is this common in other brain regions and various cell types? How does this affect spike generation and spike conductance based on axonal location? This could be useful to briefly discuss, but this comment is not a criticism or request for necessary information. This is just a curious question and something could be added only if the authors see how answering it could be beneficial to understanding LC-NE release and noradrenergic neuromodulation.

Lines 145-148 – Given that there was a sampling bias with some slices heavily enriched with FF and other slices with MP, can the authors say the dorsal-ventral levels of the FF sample biased slices? I understood that horizontal/transverse slicing was used, so some slices were more ventral than others. Were surrounding slices stained to enable precise knowledge of the location in the dorsal-ventral axis? Prior work showed that single units of narrow type were biased to be more ventral (Totah, et al. 2018. Neuron 99, 1055-1068.e6) and ventrally distributed narrow waveform units have been shown to have distinct functional roles (Li, et al. 2016. Brain Res 1641, 274-290; Hirschberg, et al. 2017. *eLife* 6, e29808). Thus, showing that FF units are also ventrally biased would help link the present work with extracellular work.

Line 204 – Given the importance of the present study in linking with extracellular recordings used in functional/behavioral studies of LC, it may be noted that Totah, et al. 2018. Neuron 99, 1055-1068.e6 found that 6% of all unit pairs had a CC consistent with a gap injection, which is very similar to the 4.3% that the authors report.

Line 209 – Do the authors mean Figure 4D instead of 4A?

Lines 220-221 – The authors may wish to note that their results are consistent with the extracellular study of CC's reporting putative gap junction mediated interactions in adult rat (Totah et al. 2018), intracellular recordings showing gap junction blockade dependent Vm coupling (Alvarez, et al. 2002. Proc National Acad Sci 99, 4032-4036), as well as anatomical studies showing gap junctions are present in adult rat LC (Bockstaele, et al. 2004. Neurochem Int 45, 421-428; Rash, J. E. et al. 2007. Neuroscience 147, 938-956).

Lines 190 – 225 – without the use of a gap junction blocker, it should be made clear that the electrical coupling is likely or putatively gap junction mediated.

Lines 411 – 414 – The description of prior work is a bit unclear when using the terms "dorsal horn" and "ventral horn" given that there is an anterior horn (which lies largely dorsal to the LC core) and a posterior horn. It would be helpful to clarify this. My understanding of Loughlin et al. 1986 is that they found: large MP cells in the ventromedial aspect of the LC core (consistent with later work by Hirschberg et al. and Li et al) and in the anterior horn of the LC (which is by definition at the dorsal edge of the LC core or even more dorsal than LC core); medium size MP cells throughout the LC core; and FF cells that were strongly anterior-posterior oriented and located on the dorsal edge of the LC core.

Loughlin also observed "small, round cells" which might be similar to the "oval" cells reported by Shimizu et al. (SHIMIZU, N., OHNISHI, S., SATOH, K. and TOHYAMA, M. 1978. Cellular Organization of Locus Coeruleus in the Rat as Studied by Golgi Method. Arch Histol Japon 41, 103-112). Did the authors happen to observe these?

Lines 496 – 499 – the finding of FF and MP specific networks of electrically coupled cells links nicely with the finding that LC pairwise correlated firing identified in extracellular recordings are cell-type specific (Totah, et al. 2018. Neuron 99, 1055-1068.e6), as are LC ensembles (Noei, et al. 2022. Proc National Acad Sci 119, e2116507119). This again is an important point because the present work serves to build a bridge between cell morphology and membrane characteristics with extracellular recordings of different unit types performed in functional/behavioral studies (Hirschberg, et al. 2017. *eLife* 6, e29808; Breton-Provencher, et al. 2022. Nature 1-7 doi:10.1038/s41586-022-04782-2).

The work is overall abbreviation heavy and the authors could write out ISD, EC, and CC instead of abbreviating them.

*Reviewer #2 (Recommendations for the authors):*

I do not think there is sufficient evidence presented here to reject the notion that local NE is released and perhaps affects LC function. Although the authors take two different current injection approaches (resulting in 1 AP or 5 AP), we simply do not know where to sample to determine if NE is playing a role. Two explanation seems just as plausible as 'no chemical transmission'. (1) The multipatch approach could just not have neurons that receive NE from the upstream neurons. (2) Even 5 AP may not release NE in a manner that would physiologically activate alpha2-AR. In this prep does blockade of alpha2 depolarize the cells? That would suggest tonic alpha2 activity that the addition of 1 or 5 APs would perhaps not elevate to the point of observation. As it stands, I would be more comfortable with the stated conclusion being that no fast-acting small molecule transmission (i.e. glutamate) was observed, rather than ruling out all chemical transmission.

Prior to eliminating the possibility of local chemical transmission entirely, it might well be worth performing this approach in coronal slices where peri-LC glutamate transmission is better maintained and the cells still firing phasically.

While the authors describe the imaging-based synapses as putative in some instances, these claims should be considered putative in every instance. Without EM or some other synaptic marker, calling these 'synapses' without the putative adjective likely exceeds what can be claimed. I would encourage the authors to ensure these limitations are clear across figures, legends, and each instance in the text. Of course, EM or other markers could be added, but that is not necessary to reach appropriately cautious conclusions.

The claims of 'high quality' and 'sufficiently healthy' LC slices should be operationally defined. What constitutes sufficiency here? Was that the real barrier to multipatch or was it more of a lack of technical and physical expertise?

Why is the n=30 on all cells that were spontaneously active? Shouldn't that number be in the 100s? Is it because the rest of the cells were hyperpolarized to prevent firing? Was there evidence of firing prior to hyperpolarization in those cells? My concern arises from the claim that 'all cells' were active when this was apparently only assayed in 30 of the ~1500 cells.

It appears there is a plotting error in Figure 4 where the trace from C8 is connected to the adjacent cell.

In the discussion, the mention of intersectional approaches should likely be expanded slightly so the meaning is clear. This section would also benefit from citing the relevant work in the LC using these approaches.

Rather than contrasting to 'classical' neurons, I think it would be more appropriate to be specific in the comparison. In other words, from hippocampal neurons rather than classical.

A few LC cell subtypes have already been established (e.g. galanin, NPY, GABA, etc.). Is it possible that either FF or MP cells align with one of these known molecular subtypes?

---

## [Author Response]

Essential revisions:1) Based on their findings, the authors argue that locus coeruleus noradrenaline neurons do not communicate through direct, excitatory synaptic transmission. However, this possibility has not been ruled out through their experiments as the horizontal slice preparation likely severed some axonal connections. The authors could try performing some experiments in coronal slices in which axonal connectivity may be somewhat more spared. Alternatively, they could moderate their claims in this aspect of the study.

We agree that coronal slice preparation may have more axons spared, and glutaminergic synaptic transmissions, if they do exist among LC neurons, may be more readily detected in coronal slices. However, we believe that these connections, if any, are sparse, and detecting these sparse connections with coronal slice preparation needs another large-scale connectivity study with extensive time and personnel investment. Thus, we chose not to pursue this line of research for this study, but we added a few sentences in the revision to discuss how we prepared the slices may affect our results (Line 503-507). Our claim was accordingly moderated as well in the revision. We could further explore this topic with the different slice preparation in our next study.

2) The authors also argue that their results support the idea that lateral inhibition through alpha2 adrenergic receptors does not occur in the LC. However, there is a wealth of data supporting the existence of lateral inhibition and the present finding are based on the stimulation of single LC-noradrenaline neurons without a more parametric analysis of different stimulation frequencies/timing. For example, it is possible, given that LC neurons often fire together, that the activity of many LC neurons at one time is required to detect lateral inhibition, as noted by the authors in the discussion. In addition, given previous work showing that the frequency and timing of spikes are important in producing and detecting lateral inhibition (see Rev. 2 comments below and Huang et al. 2007. Proc National Acad Sci 104, 1401-1406), further exploration of this question is warranted. The authors could attempt stimulation of multiple LC-noradrenergic neurons at one time (possibly using optogenetic approaches) and/or explore the parametric space more extensively. Alternatively, they could remove this section from the paper.

We thank the reviewers for their constructive comments and interpretations of the data regarding lateral inhibition. In fact, we were fully aware of the prior wealth of data supporting the existence of lateral inhibition and have discussed possible reasons for the absence of lateral inhibition in our dataset. Now both reviewers provided additional potential explanations for this absence. The most plausible explanation appears to be that α2AR-mediated lateral inhibition is a population phenomenon, which would not be readily detected at the single-cell level in in vitro conditions. As reviewers suggested, we may need to vary spike frequency and timing to identify optimal spiking parameters (or stimulating multiple LC neurons at one time) to detect this phenomenon in slices. Alternatively, we could employ other approaches (optogenetic or chemogenetic approach) to activate a group of neurons at one time to evoke this phenomenon, as a recent preprint paper demonstrated (Line 527-534). All these are excellent suggestions, but it may take more than six months to complete these experiments since we need to train another person from scratch for LC recordings (the first author graduated from the program and has left the lab). We thus chose to remove most of the data (about α2AR-mediated lateral inhibition) from the paper in the revision, as the reviewers suggested. We do plan to further explore this interesting topic in our next study.

Reviewer #1 (Recommendations for the authors):Congratulations to the authors on an extremely difficult set of experiments, stunning anatomy, and careful analyses to address important and timely questions about the LC. Really (to my knowledge) no one has dug into the anatomy like this since a few Japanese groups in 1978/1979 and Americans in mid-1980's. An update was needed – and to couple them with recordings is a truly unique data set. I have some comments and questions which I hope will help the authors to improve the manuscript and connect it better with the existing literature.One of the major strengths of the Author's work is the ability to potentially connect LC neuron types (FF and MP) with the two types of previously reported extracellular single unit. In light of that, additional characterization of a few things would be help to the field in making the connection to single units. The authors show clear data that FF cells may correspond to 'narrow units' because they have narrow AP width and have larger (albeit not significantly) AHP. Do FF have higher FR as do "narrow units"? Where were the FF neurons located in the LC core? The "narrow units" have been shown to be located throughout the core, but biased ventrally. This info would help link to existing extracellular study identifying multiple cell types.

We thank the reviewer for appreciating our work and agree that establishing the correspondence between LC neuron types (FF and MP) with previously reported two single unit types could enhance the significance of our findings. In light of a prior study in rat LC (Loughlin et al. 1986), FF cells as we defined here may correspond to the fusiform-shaped cells on the *dorsal* edge of the LC core (or the posterior horn). Given that “narrow units” are heavily biased toward the *ventral* part of the LC (Totah et al., 2018; Su and Cohen, 2022), our defined FF cells thus may not correspond to those “narrow units” reported in extracellular recordings. In addition, FF cells do not have a higher spontaneous firing rate than MPs (see Figure 2H), inconsistent with a higher firing rate of “narrow units” (Totah et al., 2018). Furthermore, there is a two-fold difference in spike width between “narrow units” and “wide units”, but the difference in spike duration (including the rise phase of first derivatives) between FF and MP is subtle despite being significant (Li et al., 2016). Therefore, there is no consistent evidence supporting that FF cells we defined here correspond to those “narrow units”. Likely, each single unit type may not be confined to a specific morphological type, or both single unit types are from the same MP type reflecting within-MP functional heterogeneity. The latter may be more likely given that MPs are a predominant LC cell type and they are more heterogeneous than FFs in terms of soma size and spatial distribution (Loughlin et al., 1986), as well as the electrophysiological properties (including AP half-width). To establish a real correspondence between cell types we defined here and those units recorded from extracellular recording may need a new systematic study. For instance, with in vivo loose patch recording, one can label these neurons via biocytin once the neurons are recorded in vivo. We have revised the Discussion regarding the correspondence between LC cell types with single units in the revision (Line 439-459).

It is a bit strange that Figure 1C and 1D come after Figure 2. It may improve the flow of the manuscript to move those panels into figure 2.

We thank the reviewer for the suggestion. We reorganized the figures (including supplement figures) as suggested in the revision.

Figure 2E – it would be helpful to have a conceptual description of what it would mean to be low x value – low y value versus high x value – high y value.

As suggested by the reviewer, we provided more details to illustrate x and y value in the revision (Line 137-144).

Figure 2F – are the rings here the number of neurons? It would be helpful to the reader if the authors could label the axis and state this in the legend.

The rings are not the number of neurons, but the dendrite orientation as a function of the dendritic length. Each radius represents dendritic length at a certain degree. We have revised the axis of Figure 2F.

Figure 2H, 2I – it would help to explain that Y axis of panel is related to branching point. It is not clear to me how to interpret this plot. it might be standard in this field but would help the general reader to have this information briefly in the Results section or figure legend to better understand the plot.

Figure 2H, 2I came from a standard Sholl analysis. Sholl analysis includes counting the number of dendritic (or axonal) intersections that occur at fixed distances from the soma in concentric circles. This analysis reveals the number of branches, branch geometry, and overall branching patterns of neurons. We included more technical details about this analysis in the revision (Figure 2 legend).

Figure 3A ("Cell 7 and Cell 2 is electrically coupled") – "is" should be "are".

Corrected.

Figure 7C – This plot in C is confusing. The methods state that a horizontal (transverse) plane was used, therefore, ventral and dorsal should be medial and lateral. Can the Authors clarify this?

The plot in Figure 7C is a sagittal view of LC, not a horizontal view, and the proposed LC subdivisions and predominant cell types for each subdivision were adopted from Loughlin et al. 1986 with a slight modification. As mentioned in our response to a question below, estimating in vivo anatomical locations of two cell types (MP and FF) and identifying which LC subdivision they may be predominantly located at is difficult (e.g., by mapping each cell’s soma location to the Allen Brain Atlas), and therefore we could not generate an LC schematic diagram based on our real data. We have revised the Figure 7 legend to make this point clearer.

Line 78 – strike "the".

Corrected.

Line 94 – Can the authors comment on Why the RMP is so different from what was reported in earlier work (see Supplementary Table 5 in Chandler, Gao, and Waterhouse. 2014. Proc National Acad Sci 111, 6816-682)?

We reported the RMP values without correcting junction potentials, and these values are very similar to those reported in several previous reports (Ballantyne et al., 2004; Kuo et al., 2020). The difference between ours and Chandler, et al. 2014 may be due to the fact that RMP values in Chandler, et al. 2014 have been corrected with junction potentials. We added a statement to reflect this point in the revised Method section (Line 787 or 788).

Lines 132-134 – The fact that the axon originated from different locations on FF versus MP cells is interesting. Is this common in other brain regions and various cell types? How does this affect spike generation and spike conductance based on axonal location? This could be useful to briefly discuss, but this comment is not a criticism or request for necessary information. This is just a curious question and something could be added only if the authors see how answering it could be beneficial to understanding LC-NE release and noradrenergic neuromodulation.

Cortical pyramidal neurons (PNs) in general have the axon originating from the soma, but a small subset of PNs could have the axon originating from the dendrites (Hamada et al., 2016). The dendrite-originated axon is more common in GABAergic interneurons (particularly VIP and SST-expressing interneurons) and aminergic neurons (Hausser et al., 1995). Therefore, it is common that neurons can have the axon originating from different locations, particularly for aminergic neurons. Given that action potential initiates from the axonal initial segment, the location where the axon originates will affect how synaptic inputs are integrated and converted into the outputs of the neurons. For those cells such as FF whose axons originate from the dendrite, those synapses made on the axon-bearing dendrite are in an electrotonically privileged position in evoking action potentials compared to synapses made on other dendrites. Therefore, MP and FF may have different synaptic integration mechanisms. We added a sentence to briefly illustrate this point in the revision (Line 170-172).

Lines 145-148 – Given that there was a sampling bias with some slices heavily enriched with FF and other slices with MP, can the authors say the dorsal-ventral levels of the FF sample biased slices? I understood that horizontal/transverse slicing was used, so some slices were more ventral than others. Were surrounding slices stained to enable precise knowledge of the location in the dorsal-ventral axis? Prior work showed that single units of narrow type were biased to be more ventral (Totah, et al. 2018. Neuron 99, 1055-1068.e6) and ventrally distributed narrow waveform units have been shown to have distinct functional roles (Li, et al. 2016. Brain Res 1641, 274-290; Hirschberg, et al. 2017. eLife 6, e29808). Thus, showing that FF units are also ventrally biased would help link the present work with extracellular work.

With horizontal slicing (2-3 slices in total with 200 µm thickness), the first slice tends to include the posterior horn of the LC and the dorsal part of the LC core, while the last one or two slices are more likely to include the ventral part of the LC core as well as the anterior horn (based on the Allen Brain Atlas). If the cellular organization of mouse LC is the same as that of rat LC (illustrated by Laughlin et al.1986), the first slice is expected to bias toward FF cells (FF sample biased slices), while the last one or two slices bias toward MP cells (MP sample biased slices). Unfortunately, we lost track of the slice order (the first collected vs the last collected from each animal) of all recorded slices, and thus were unable to confirm if those FF sample biased slices were collected earlier than those MP biased slices. Nevertheless, we noticed that in each slice FFs in general appear to be closer to the 4^th^ ventricle (more dorsal) than MPs, suggesting FFs in mice may indeed be dorsally biased, similar to rat LC (Loughlin et al. 1986). In terms of the correspondence of our intracellular slice work with prior in vivo single units, please see our response to the reviewer’s first question.

Line 204 – Given the importance of the present study in linking with extracellular recordings used in functional/behavioral studies of LC, it may be noted that Totah, et al. 2018. Neuron 99, 1055-1068.e6 found that 6% of all unit pairs had a CC consistent with a gap injection, which is very similar to the 4.3% that the authors report.

As suggested, a sentence was added in the Discussion (Line 541) to highlight the consistency of our results with prior work.

Line 209 – Do the authors mean Figure 4D instead of 4A?

Yes. This was a typo and was corrected in the revision.

Lines 220-221 – The authors may wish to note that their results are consistent with the extracellular study of CC's reporting putative gap junction mediated interactions in adult rat (Totah et al. 2018), intracellular recordings showing gap junction blockade dependent Vm coupling (Alvarez, et al. 2002. Proc National Acad Sci 99, 4032-4036), as well as anatomical studies showing gap junctions are present in adult rat LC (Bockstaele, et al. 2004. Neurochem Int 45, 421-428; Rash, J. E. et al. 2007. Neuroscience 147, 938-956).

We thank the reviewer for the suggestion. We cited these references and added a sentence to briefly discuss how our results align with prior work in the revision (Line 541-544).

Lines 190 – 225 – without the use of a gap junction blocker, it should be made clear that the electrical coupling is likely or putatively gap junction mediated.

As suggested, we revised our manuscript to make this point clear.

Lines 411 – 414 – The description of prior work is a bit unclear when using the terms "dorsal horn" and "ventral horn" given that there is an anterior horn (which lies largely dorsal to the LC core) and a posterior horn. It would be helpful to clarify this. My understanding of Loughlin et al. 1986 is that they found: large MP cells in the ventromedial aspect of the LC core (consistent with later work by Hirschberg et al. and Li et al) and in the anterior horn of the LC (which is by definition at the dorsal edge of the LC core or even more dorsal than LC core); medium size MP cells throughout the LC core; and FF cells that were strongly anterior-posterior oriented and located on the dorsal edge of the LC core.

We agree with the reviewer that our description of this prior work was a bit simplified and incomplete. We provided a more detailed description of this prior work in the revision (Line 419-421).

Loughlin also observed "small, round cells" which might be similar to the "oval" cells reported by Shimizu et al. (SHIMIZU, N., OHNISHI, S., SATOH, K. and TOHYAMA, M. 1978. Cellular Organization of Locus Coeruleus in the Rat as Studied by Golgi Method. Arch Histol Japon 41, 103-112). Did the authors happen to observe these?

These small, round cells may be GABAergic interneurons in the LC (Shimizu et al., 1979; Léger and Hernandez-Nicaise, 1980; Aston-Jones et al., 2004; Breton-Provencher and Sur, 2019) or central gray cells, as suggested by the same authors. These cells thus may not be included in our study (not labeled by the *Dbh-cre* line). However, we did see one or two cells of a similar profile out of ~700 cells recorded, which may be due to the fact that we accidentally patched a few unlabeled neurons. These cells have different firing patterns from most LC cells we recorded and were not included in the analysis.

Lines 496 – 499 – the finding of FF and MP specific networks of electrically coupled cells links nicely with the finding that LC pairwise correlated firing identified in extracellular recordings are cell-type specific (Totah, et al. 2018. Neuron 99, 1055-1068.e6), as are LC ensembles (Noei, et al. 2022. Proc National Acad Sci 119, e2116507119). This again is an important point because the present work serves to build a bridge between cell morphology and membrane characteristics with extracellular recordings of different unit types performed in functional/behavioral studies (Hirschberg, et al. 2017. eLife 6, e29808; Breton-Provencher, et al. 2022. Nature 1-7 doi:10.1038/s41586-022-04782-2).

We appreciate the reviewer’s insightful comments. We added a few sentences in the Discussion to connect our work to prior findings from extracellular recordings (Line 573-576).

The work is overall abbreviation heavy and the authors could write out ISD, EC, and CC instead of abbreviating them.

We thank the reviewer for the suggestion. These abbreviations were spelled out across the manuscript in the revision.

Reviewer #2 (Recommendations for the authors):I do not think there is sufficient evidence presented here to reject the notion that local NE is released and perhaps affects LC function. Although the authors take two different current injection approaches (resulting in 1 AP or 5 AP), we simply do not know where to sample to determine if NE is playing a role. Two explanation seems just as plausible as 'no chemical transmission'. (1) The multipatch approach could just not have neurons that receive NE from the upstream neurons. (2) Even 5 AP may not release NE in a manner that would physiologically activate alpha2-AR. In this prep does blockade of alpha2 depolarize the cells? That would suggest tonic alpha2 activity that the addition of 1 or 5 APs would perhaps not elevate to the point of observation. As it stands, I would be more comfortable with the stated conclusion being that no fast-acting small molecule transmission (i.e. glutamate) was observed, rather than ruling out all chemical transmission.

We appreciate the reviewer’s constructive comments and suggestions. Please see our response to Essential Revisions.

Prior to eliminating the possibility of local chemical transmission entirely, it might well be worth performing this approach in coronal slices where peri-LC glutamate transmission is better maintained and the cells still firing phasically.

This is a good point. Please see our response to Essential Revisions.

While the authors describe the imaging-based synapses as putative in some instances, these claims should be considered putative in every instance. Without EM or some other synaptic marker, calling these 'synapses' without the putative adjective likely exceeds what can be claimed. I would encourage the authors to ensure these limitations are clear across figures, legends, and each instance in the text. Of course, EM or other markers could be added, but that is not necessary to reach appropriately cautious conclusions.

This is a valid point. We revised the manuscript as suggested.

The claims of 'high quality' and 'sufficiently healthy' LC slices should be operationally defined. What constitutes sufficiency here? Was that the real barrier to multipatch or was it more of a lack of technical and physical expertise?

LC is a small nucleus containing only ~1,500 neurons in mice in each hemisphere. With a regular slicing procedure (using regular ACSF), each slice has sparse alive LC neurons and performing multi-cell recordings is infeasible. Only with the NMDG-based slicing protocol could we obtain the slices with a sufficient number of healthy cells (judged by their RMP) for multipatch recording (“high-quality” slices). Here “sufficient” is a modifier for LC neurons (meaning a sufficient number of neurons), not for “healthy”. We revised the text to avoid ambiguity (Line 85-90).

Why is the n=30 on all cells that were spontaneously active? Shouldn't that number be in the 100s? Is it because the rest of the cells were hyperpolarized to prevent firing? Was there evidence of firing prior to hyperpolarization in those cells? My concern arises from the claim that 'all cells' were active when this was apparently only assayed in 30 of the ~1500 cells.

All cells were spontaneously active at rest (this was noticed upon entering whole-cell), consistent with prior observations (Andrzejewski et al., 2001; Ballantyne et al., 2004; Kuo et al., 2020). To suppress their spontaneous firing, most cells were hyperpolarized to allow for testing connections (see Method section). Only about 30 cells were recorded for a certain period of time at rest (with zero current injection) to allow for characterizing their firing rate at rest.

It appears there is a plotting error in Figure 4 where the trace from C8 is connected to the adjacent cell.

The error was corrected.

In the discussion, the mention of intersectional approaches should likely be expanded slightly so the meaning is clear. This section would also benefit from citing the relevant work in the LC using these approaches.

We revised the manuscript as suggested. Please see Line 479-489.

Rather than contrasting to 'classical' neurons, I think it would be more appropriate to be specific in the comparison. In other words, from hippocampal neurons rather than classical.

We revised the manuscript as suggested.

A few LC cell subtypes have already been established (e.g. galanin, NPY, GABA, etc.). Is it possible that either FF or MP cells align with one of these known molecular subtypes?

This is a good question. Based on immunostaining of rat LC, only a small subset of noradrenergic neurons (23%) in LC are NPY-expressing and their soma shape could be either multipolar or fusiform; almost all LC noradrenergic neurons (82%) are galanin-positive, and their soma shape could be either multipolar or fusiform as well (Melander et al., 1986; Holets et al., 1988). Therefore, it seems that the molecular subtype defined by the expression of NPY or galanin is not confined to a specific morphological type. A brief discussion on these two genes for labeling LC neurons was added in the revision (Line 475-479).

GABAergic neurons in the LC intermingle with and surround NA neurons (preferentially located in the anterior and ventral part of the LC area) and do not express *Dbh* (non-noradrenergic neurons) (Aston-Jones et al., 2004; Breton-Provencher and Sur, 2019). Therefore, they were not included in this study (not labeled by the *Dbh-cre* line; Line 93-96).